# Reproducing an extreme flood with uncertain post-event information

**Diana Fuentes-Andino[1,2], Keith Beven[1,3], Sven Halldin[1,2], Chong-Yu Xu[1,4], José Eduardo Reynolds[1,2] and Giuliano Di Baldassarre[1,2]**

[1]{Department of Earth Sciences, Uppsala University, Villavägen 16, SE–752 36 Uppsala, Sweden}

[2]{Centre for Natural Disaster Science (CNDS), Uppsala University, Uppsala, Sweden}

[3]{Lancaster Environment Centre, Lancaster University, Lancaster LA1 4YQ, UK}

[4]{Department of Geosciences, University of Oslo, P O Box 1047, Blindern, NO–0316, Oslo, Norway}

Correspondence to: D. Fuentes (diana.fuentes@geo.uu.se)

## Abstract

Studies for the prevention and mitigation of floods require information on discharge and extent of inundation, commonly unavailable or uncertain, especially during extreme events. This study was initiated by the devastating flood in Tegucigalpa, the capital of Honduras, when Hurricane Mitch struck the city. In this study we hypothesised that it is possible to estimate, in a trustworthy way considering large data uncertainties, this extreme 1998 flood discharge and the extent of the inundations that followed, from a combination of models and post-event measured data. Post-event data collected in 2000 and 2001 were used to estimate discharge peaks, times of peak and high-water marks. These data were used in combination with rain data from two gauges to drive and constrain a combination of well-known modelling tools: TOPMODEL, Muskingum-Cunge-Todini routing, and the LISFLOOD-FP hydraulic model. Simulations were performed within the GLUE uncertainty-analysis framework. The model combination predicted peak discharge, times of peaks and more than 90% of the observed high-water marks within the uncertainty bounds of the evaluation data. This allowed an inundation likelihood map to be produced. Observed high-water marks could not be reproduced at a few locations on the floodplain. Identification of these locations are useful to improve model set-up, model structure or post-event data-estimation methods. Rainfall data

were of central importance in simulating the times of peak and results would be improved by a better spatial assessment of rainfall, *e.g.* from radar data or a denser rain-gauge network. Our study demonstrated that it was possible, considering the uncertainty in the post-event data, to reasonably reproduce the extreme Mitch flood in Tegucigalpa in spite of no hydrometric

gauging during the event. The method proposed here can be part of a Bayesian framework in which more events can be added into the analysis as they become available.

**Keywords:** post-event measured data, extreme floods, rainfall-runoff and hydraulic model combination, uncertainty analysis, ungauged basins.

**1    Introduction**

Losses caused by natural hazards have a significant impact on the world economy, and floods account for around half of all disasters globally (UN/ISDR, 2016). Prevention and mitigation of floods require information on discharge and extent of inundation. Such information is commonly unavailable or uncertain, especially during extreme events when gauging

equipment becomes insufficient or is lacking. Data scarcity is further aggravated in developing countries with weak infrastructure.

Nearly 11 000 people were killed in Central America during Hurricane Mitch because of extreme flooding, an estimated 2.7 million lost their homes and flood damages were estimated to more than 6 billion USD (McCown *et al*., 1999). This study was initiated by the flood in

Tegucigalpa, the capital city of Honduras, on 30–31 October 1998 when Mitch struck the city. The estimated 500–year return period rainfall produced by Mitch (JICA, 2002) caused significant damage to Tegucigalpa, where one thousand casualties were reported and approximately 40% of its capital stock was damaged (Angel *et al*., 2004; JICA, 2002). In addition to these calamities, much of Honduras' hydrological archives were swept away from

their premises at SANAA (Servicio Autónomo Nacional de Acueductos y Alcantarillados) which was located close to the main channel of the upper Choluteca River.

Simulations of water-level dynamics caused by disastrous events are needed for preparedness, to produce flood-inundation maps useful for urban planning and to prioritise investments (Pappenberger *et al*., 2006; Schanze, 2006). Such simulations are also relevant to better

comprehend the hydraulic mechanism of large flood events in order to improve model structure (Beven *et al*., 2011; Jarrett, 1990). However, given that simulations of extreme

floods are generally associated with limited data availability and large uncertainties, the question arises as to whether it is possible to achieve simulations that can be useful for contingency planning and prevention.

When hydrometric measurements of discharge and water levels during an event are lacking or highly inaccurate, such information may be inferred from post-event surveys. These can be done through eye-witness accounts and field campaigns (Brandimarte and Di Baldassarre, 2012; Ciervo *et al*., 2015; Gaume and Borga, 2008; Horritt *et al*., 2010; JICA, 2002; Smith *et al*., 2002), sometimes in combination with additional methods such as search into historical documentation and paleo-flood techniques (Mård Karlsson *et al*., 2009; Smith *et al*., 2012; Valyrakis *et al*., 2015). Such surveys have been useful to estimate hydrometric data of the floods. Pictures and movies can be used to identify locations, flow type, depth, flow velocity and discharge at the time they were taken (*e.g.* Ciervo *et al*., 2015; Le Boursicaud *et al*., 2016). Post-event information of channel topography and maximum water level can be used to estimate maximum peak discharge (Dalrymple and Benson, 1968; Matthai, 1968).

Post-event-estimated maximum peak discharge can be used to produce probabilistic regional envelope curves (Castellarin, 2007; Gaume *et al*., 2009) and discharge series for flood-frequency analysis (Cœur and Lang, 2008). These provide design-flood estimates used for inundation mapping (*e.g.* Brandimarte and Di Baldassarre, 2012). However, an assessment of flood development in time is required for early-warning systems (Schanze, 2006). The development of a flood in time can be obtained through a strategically planned post-event survey of peak discharge and the associated time of the peak (*e.g.* Delrieu *et al*., 2005). Detailed hydrographs can also be obtained from rainfall time series in conjunction with post-event hydrometric data, by the use of a rainfall-runoff model (RRM). A RRM in turn can be coupled with a hydraulic model to estimate the water-level development along a floodplain (Bonnifait *et al*., 2009; JICA, 2002; Montanari *et al*., 2009; Pappenberger *et al*., 2005a). Results from hydraulic models can be validated against post-event-estimated peak discharge, time of the peak, maximum water-level and flood-extent data (*e.g.* Bonnifait *et al*., 2009; Brandimarte and Di Baldassarre, 2012; Horritt *et al*., 2010).

Post-event data have been used with deterministic calibration within hydraulic models (*e.g.* Horritt *et al*., 2010; JICA, 2002), and for coupling RRM with hydraulic models (*e.g.* Ciervo *et al*., 2015). Using post-event data, Bonnifait *et al*. (2009) present a multi-variable assessment to find a group of best parameter sets for the TOPMODEL RRM and a 1D hydraulic model.

Borga *et al*. (2008) and Pappenberger *et al*. (2006) suggest that post-event data should be used within an uncertainty-analysis framework given their large uncertainties. Di Baldassarre *et al*. (2010) discussed the advantages of distributed uncertainty mapping, as first proposed by Romanowicz and Beven (1998), in comparison with deterministic mapping. Uncertainty analysis techniques have been used to account for uncertainty in hydraulic models (Aronica *et al*., 1998; Bozzi *et al*., 2015; Brandimarte and Di Baldassarre, 2012; Pappenberger *et al*., 2005a, 2007) and for the coupling of a RRM with a hydraulic model (Montanari *et al*., 2009; Pappenberger *et al*., 2005a) using event-measured data.

Uncertainty-analysis techniques account for possible errors involved in the modelling process, *e.g.* errors in model parameters and input data, due to lack of knowledge of their true values, spatio-temporal variability, or inaccurate estimation, and errors related to limited knowledge of the behaviour of the real system, *i.e.* epistemic uncertainty (Beven, 2016, 2009). Thus in uncertainty-analysis techniques, uncertainties can be associated with several sources that interact among them, in which each interaction is associated with a likelihood dependent on how well it fits the observations. The formal Bayesian approach is a widely used method for uncertainty analysis, with different setups available (*e.g.* Smith and Roberts, 1993). Bayesian techniques have been commonly applied in hydraulic and hydrological modelling (*e.g.* Hall *et al*., 2011; Renard *et al*., 2008) and can be used within a global sensitivity analysis (see summaries by Iooss and Lemaître (2015) and Sarrazin *et al*. (2016)) to assess the effect of each source of uncertainty on the output (*e.g.* Abily *et al*., 2016). An informal Bayesian approach is the Generalised Likelihood Uncertainty Estimation (GLUE) framework (Beven and Binley, 1992), which differs in the way likelihood is defined and in that it does not require a prior knowledge on the correlations or distributions of the parameter errors, yet with GLUE it is possible to get posterior information in the parameter combinations. In this study we hypothesise that it is possible to reasonably estimate, considering the large uncertainties in the observations, the extreme 1998 flood discharge in Tegucigalpa and the extent of the inundations that followed, from a combination of models and post-event data. We are aware of works that use the combination of hydraulic models and RRMs to assess flood dynamics or others that use post-event data to calibrate either RRMs or hydraulic models, both deterministic and through uncertainty analyses. We are not aware of any previous study combining a RRM, hydraulic modelling, and post-event data within an uncertainty analysis framework to prove that reasonable estimation of an extreme flood is possible when hydrometric data are lacking. The methodology suggested in this paper integrates

TOPMODEL (Beven and Kirkby, 1979; Kirkby, 1997), Muskingum-Cunge-Todini (MCT) (Todini, 2007) routing, and the LISFLOOD-FP (Neal *et al*., 2012a)*et al* hydraulic modelling tool in a GLUE framework.

## 2  Study area and data

### 2.1  Area description

The study area was the floodplain at Tegucigalpa City, approximately 13 km of river length downstream of the upper part of the Choluteca River catchment. The area draining to the floodplain is around 811 km$^2$ and is composed of five sub-catchments: Grande River (448 km$^2$), Guacerique River (243 km$^2$), Chiquito River (71 km$^2$), Salada Creek (25 km$^2$) and Las Lomas Creek (12 km$^2$) (Fig. 1). Rainfall in the region is affected by high hurricane recurrence (Alvarado and Alfaro, 2003; Strobl, 2009) and convective activity. These two features in combination with the mountainous nature of the terrain (Amador *et al*., 2006) might lead to a high spatial variation of rainfall. Westerberg *et al*. (2010) found that daily precipitation has a high spatial variability and that bias in the estimations are likely due to insufficient gauge stations to measure in space and at different elevations. The land use and geology are relatively uniform in all sub-catchments. The land use is mainly composed of sparse coniferous forest at higher elevation lands; fallow, pastures and urbanised areas in the low land (CIAT, 2007). The geology at the surface is mainly composed of tuff and limestone to a minor degree; the superficial aquifer is classified as poor to moderately productive (ING, 1996). The average basin slope estimated in the Grande River, Guacerique River, Chiquito River, Salada Creek and Las Lomas Creek sub-catchments is 19.5, 18, 25, 17.5 and 11 % respectively. Two reservoirs operated by SANAA are established upstream the Tegucigalpa floodplain: the Concepción reservoir, located at Grande River sub-catchment, and Los Laureles reservoir, located at Guacerique River sub-catchment (Fig. 1).

## 2.2 Data

### 2.2.1 Topography

An airborne light-detection and ranging (LIDAR) survey in Tegucigalpa was conducted in 2000 by the University of Texas in cooperation with the U.S. Geological Survey (USGS) during their survey in Honduras in response to Hurricane Mitch (Mastin, 2002). They generated a 1.5 m cell-resolution digital-terrain model (DTM) with an estimated vertical accuracy of 0.14 m (Fig. 2). These LIDAR data were used by Haile and Rientjes (2005) to investigate the effect of a Digital Elevation Model (DEM) resolution on simulated flood extension using the SOBEK modelling tool. In 2001, JICA (2002) also conducted a topographic field survey as part of a flood/landslide-mitigation master plan and a total of 99 cross-sections along the rivers in the floodplain at Tegucigalpa, surveyed at intervals of approximately 100 m, were used in this study (Fig. 2). In addition, orthographic pictures were taken at Tegucigalpa city by JICA (2002).

The topography of the Tegucigalpa floodplain upstream sub-catchments was available from the 90–m spatial resolution Shuttle Radar Topography Mission (SRTM) data described by Reuter *et al*. (2007) (Fig. 1).

### 2.2.2 Precipitation

Upstream the Tegucigalpa floodplain, two stations measured hourly rainfall during the Mitch event (Fig. 1 and 3). One of the stations is operated by Servicio Meteorológico Nacional (SMN, national weather service) and the other by the Universidad Nacional Autónoma de Honduras (UNAH).

### 2.2.3 Discharge

Discharge at three locations was estimated post-event by Smith *et al*. (2002) using the standard USGS techniques by Benson and Dalrymple (1967). The peaks at Chiquito River and Grande River (points 1 and 2 in Fig. 1 and Table 1) were estimated using the width-contraction analysis that uses the continuity and energy equations between a cross-section approaching the contraction section under a bridge (Matthai, 1968). The peak at Choluteca (point 3) was estimated using the slope-area analysis, in which discharge is computed on the basis of the uniform-flow equation involving channel geometry, high water marks, and roughness coefficients (Dalrymple and Benson, 1968). The measurements of discharge using

the width-contraction analysis and the slope-area analysis can be associated with 25% error for unfavourable field-data conditions (Benson and Dalrymple, 1967), but up to 100% overestimation might be associated with the slope-area analysis for slopes greater than 0. 2% (Jarrett, 1987).

A deterministic reproduction of the flood produced by hurricane Mitch was done by JICA (2002) by setting a rainfall-runoff analysis, a linear reservoir model driven with hourly rainfall data from the SMN station. The produced hydrograph was used as input for the 1D Mike 11 modelling tool (DHI, 2000) for unsteady flow conditions. In addition to the flood extent (Fig. 2), JICA (2002) reported the maximum peak discharge at the points 4, 5 and 7 in Figs. 1, 2

and Table 1.

Controlled flow release through the spillway at the Concepción reservoir was conducted and recorded by SANAA during the Mitch event (Fig. 3). The outflow over Los Laureles dam was not recorded. However, SANAA reported that its gate was overtopped at 22:30 on 30 October, reaching a maximum of approximately 1 200 $m^3s^{-1}$ (JICA, 2002; Smith *et al*., 2002).

Peak times in Table 1 except at point 5, were obtained by interviewing witnesses. The time of the peak at point 5 was estimated by propagating the peak reported at los Laureles reservoir.

### 2.2.4  Maximum water levels

High-water marks during the Mitch flood were surveyed post-event by JICA (2002); the data were obtained by interviewing residents who experienced the event. The survey was carried

out at the same locations where the topographic cross-sections were made (Fig. 2).

## 3    Method

### 3.1    Consistency in the post-event measured data

An inspection of the consistency of the data was done prior to the analysis. The inspection

was done by plotting the maximum water-level profile to detect possible outlier. The consistency in timing and magnitude along the river network for the post-event maximum peak discharge was also checked. The flood-wave peak and time of the peak were expected to be larger and later downstream the river confluences, respectively.

### 3.2 Uncertainty and evaluation function

To quantify the propagation of uncertainty, the GLUE method was used. The assumptions of more formal statistical approaches, can not be justified in data-scarce cases with high epistemic uncertainties. Within the GLUE methodology, parameter sets were generated using a Monte Carlo technique, assuming a uniform prior distribution of the parameters.

Behavioural parameter sets, those that perform well in predicting the observations, were selected using a likelihood measure that reflected the performance of individual simulations with respect to one or several evaluation variables ($o_i$). Likelihoods were inferred by using the degree of belief ($d_i$) of a trapezoidal fuzzy membership function (Fig. 4), which shape was chosen to account for uncertainties in the post-event estimated values, which are not considered crisp estimations. Thus the degree of belief for a difference smaller than $a$ between the simulated and post-event estimated values is equal to one and it declines linearly to zero for differences larger than $b$.

### 3.3 Modelling framework

The dynamic of the water level along the river channel and floodplain was reproduced with the sub-grid channel formulation of the LISFLOOD-FP hydrodynamic model (Neal *et al.*, 2012a)*et al*. The model requires flow hydrographs as upstream boundary condition, which were generated using the RRM TOPMODEL (Beven and Kirkby, 1979; Kirkby, 1997) as in Fuentes-Andino *et al*., (2017) (Appendix A) together with the Muskingum-Cunge-Todini (MCT) flood-routing approach (Todini, 2007) (Appendix B). A scheme of the modelling framework is shown in Fig. 5.

### 3.4 Representative hydrographs for the upstream boundary condition

Topographic information is a basis to set-up TOPMODEL, which was one reason to select it in our mountainous catchment. Additionally, the version used here (Fuentes-Andino *et al*., 2017) has shown to improve model prediction by considering the uncertainty associated with the spatial averaged estimation of rainfall. The mass-conservative version of the Muskingun-Cunge routing, the MCT, was incorporated to consider the sudden release of water from the Concepción reservoir, and it was chosen since a more complex routing could not be applied

given the lack of data at the upstream area of the floodplain. The effect of Los Laureles dam on simulating the hydrograph of the Guacerique River sub-catchment was assumed to be negligible since the dam was overtopped much before the most intensive period of the storm.

The TOPMODEL and MCT combination assumes slope-dependent variable velocity at hillslope, constant velocity at normal channel and a variable velocity (according to the diffusive wave model of the MCT) at the main channel (which length was estimated having a minimum drainage area equal to 65 km$^2$). For each sub-catchment, the main channel was sub-divided in reaches of approximately 2.5 km to execute the MCT routing. For the MCT routing at Grande River, the inflow for the most upstream reach was set equal to the outflow hydrograph from the reservoir, and for other sub-catchments, to be equal to the hydrograph draining to that reach using TOPMODEL. For the subsequent reaches, this inflow was estimated as the sum of the outflow from the MCT routing at the immediate upstream reach and the hydrograph produced by TOPMODEL on the area draining to that reach (excluding the area draining to the upstream reaches). The modelling time step was equal to five minutes, smaller than the estimated travel time of the flood wave along the reach, as required by the MCT routing.

For the TOPMODEL, a network width function for each reach was created using topography from the SRTM raster. Only two rain-gauge stations were available, which made it difficult to infer the spatial distribution of rainfall. However rainfall registered at the two stations was similar, thus rainfall was assumed spatially uniform and estimated as the average of the two time series. Given the large magnitude of the event, it was expected to be associated with little spatial variation.

Uncertainty in rainfall input was taken into account by a multiplier ($R$), in addition, uncertainty of six model parameters was considered: the rate of decline of transmissivity ($m$), horizontal transmissivity ($T_o$), time constant ($t_d$), land-use coefficient ($l_u$), flood-wave celerity ($v_c$) and maximum soil infiltration rate ($i_{max}$) (Appendix A). The MCT method required information of the river slope and the geometry of the cross-sections (Appendix B). The former was approximated from SRTM data, while the latter was inferred here as a function of discharge using the Manning equation for a wide parabolic channel as in Tewolde and Smithers (2007), with channel roughness coefficient ($n_{cu}$) assumed uniform along all the reaches to make the modelling system simple and in view of the lack of data to constrain localised values.

All parameters were sampled from uniform distributions with ranges considered large but possible in the literature (Table 2) and each generated parameter set was used to simulate Chiquito, Grande and Guacerique River sub-catchments (outlets at points 1, 2 and 5 in Figs. 1 and 2 and Table1). A stopping criterion as in Pappenberger *et al*. (2005b) was used to decide the number of simulations required. For every 500 behavioural simulations added, a cumulative distribution function (CDF) of the predicted peak discharge and one of the time of the peak were estimated (evaluation variables, see section 3.4.1). These estimated CDFs were compared with the previous one and the number of runs was considered sufficient when the addition of behavioural simulations did not change the CDF significantly (*i.e.* P < 0.05) using the Kuiper (1960) statistic test (Appendix C). This statistical test was considered suitable since it is sensitive to changes in the tail and to the median values of the distribution, therefore it makes sure that the distributions did not change along the whole range of values.

Las Lomas Creek and Salada Creek (points 8 and 9 in Figs. 1 and 2) did not have data to constrain the simulations and, by proximity, the behavioural parameters found at both Grande and Chiquito were used to simulate them. This is expected to not largely affect the system as the contributing areas for Las Lomas and Salada Creek are relatively small in comparison to the three sub-catchments where post-event data were available (Fig. 1). In addition, these two areas were smaller than the threshold drainage area for applying MCT, therefore only parameters from TOPMODEL were transferred to those sub-catchments.

### 3.4.1 Output evaluation

To decide on behavioural hydrographs for Chiquito, Guacerique and Grande River sub-catchments the maximum peak and time of peak post-event observations , together with their associated uncertainty, were used (refer to points 1, 2 and 5 in Table 1). The assumed uncertainty range was $b = a = \pm50\%$ of the peak flow for the peak magnitude and $b = a = \pm2.5$ hours for the time of peak (Fig. 4). For the evaluation of the hydrographs, $a$ was set equal to $b$, thus every hydrograph within the uncertainty bounds was considered behavioural and to have equal degree of belief. The uncertainty in peak discharge at points 1 and 2 was chosen considering, and assumed larger than the value suggested at Benson and Dalrymple (1967). The discharge at point 5, although it was estimated by running a RRM by JICA (2002), was considered reliable for calibration since its magnitude was similar to the maximum peak outflow measured at Los Laureles dam, located in the same river and with nearly equal contributing upstream areas as in point 5 (Fig.1). It was expected that 50% of the

flow uncertainty, as well as for points 1 and 2, was also reasonable at point 5. All times of peak came from the same source, *i.e.* witness accounts, and there was no additional information on their uncertainties, thus we allowed up to 2.5 hours uncertainty considering that the survey was carried two years after the event and because of the expected difficulties

in witnesses identifying the exact times when the peak occurred.

To reduce computational costs and avoid redundancy, 100 representative hydrographs (class hydrographs) were obtained for each sub-catchment by clustering the full behavioural ensemble. Clustering was done using the K-means flat algorithm also called Lloyd's algorithm, originally developed by Lloyd (2006), described in Madhulatha (2012) with tool

available for use at Mathworks (2011). Following the K-means algorithm, the number of groups (K) to cluster an ensemble of data (here the behavioural hydrographs) were defined (here equal to 100). Then, a number of K hydrographs were randomly chosen from the ensemble to represent the clusters centroids. Each of the hydrographs in the ensemble was assigned to one of the centroid hydrographs according to the smallest distance, here taken as

the sum of the absolute differences between hydrographs. Subsequently the centroid for each of the cluster was replaced by the average hydrograph within each cluster and then each hydrograph in the ensemble was assigned to the new centroid found. The procedure of moving centroids and assigning hydrographs to new centroids is repeated until there is no change in the clusters. To consider the extreme cases, the hydrograph from each cluster with

the largest sum of the distances to all other centroid hydrographs was chosen.

### 3.5 Flood-wave propagation

The LISFLOOD-FP was used to propagate the flood waves along the channels and across the flood plain. Here the sub-grid channel formulation after Neal *et al*. (2012a)*et al* was used, where the floodplain and the channel have a 2D square grid representation and flow is

conveyed using the local inertia formulation (de Almeida *et al*., 2012). Thus, the continuity equation (Eq. 1) and a simplified version of the momentum equation (where the convective-acceleration term was assumed negligible) (Eq. 2) were used to keep the continuity of mass and momentum in each cell and between cells respectively.

$$\frac{\partial h}{\partial t} + \frac{\partial Q_x}{\partial x} + \frac{\partial Q_y}{\partial y} = 0, \tag{1}$$

$$\frac{\partial Q_x}{\partial t} + gA\left(S_x + \frac{n^2 Q_x |Q_x|}{(R^{4/3})A^2}\right) = 0, \tag{2a}$$

$$\frac{\partial Q_y}{\partial t} + gA\left(S_y + \frac{n^2 Q_y |Q_y|}{(R^{4/3})A^2}\right) = 0 \, , \qquad\qquad\qquad (2b)$$

where $Q_x$ and $Q_y$, $S_x$ and $S_y$ are the volumetric flow rates and the slopes respectively in the x and y directions, $h$ the water depth, $t$ time, $A$ the cross-sectional area of flow, $g$ gravity, $n$ the Manning's coefficient and $R$ the hydraulic radius, taken as the cell cross-section area divided by the wetted perimeter.

Equations 1 and 2 are solved using an explicit forward difference scheme on a staggered grid (Bates *et al*., 2010) which requires fewer numerical operations (about an order of magnitude ) than a full 2D dynamic model (Neal *et al*., 2012b). The former numerical procedure was computationally more efficient than the latter and therefore more suitable for uncertainty analysis. In addition, the model-grid representation made it possible to obtain the discharge and water-level time series output at any grid along the channel or floodplain.

The basic input data for the LISFLOOD-FP are topography, hydrographs at the upstream boundary conditions, a downstream boundary condition and Manning roughness coefficients. To use as topographic input to the model, the LIDAR data were aggregated to 21 meter cell resolution, as a trade-off between high resolution and the speed of simulations. The surveyed cross-sections and orthographic pictures from JICA (2002) were used to define channel depth and width respectively. Test simulations of this event were performed within the HEC-RAS one-dimensional hydraulic model (Brunner, 2001) considering the topography of the bridges, and preliminary results showed that bridges had a negligible effect on the overall flood profile. Thus, the geometry of bridges in the LISFLOOD-FP implementation was neglected by assuming a limited and localised impact on flood levels as in *e.g.* Castellarin *et al*. (2009), especially since the calibration data are associated with large uncertainties so that the localised effect of structures is not possible to detect (Fewtrell *et al*., 2011).

Uncertainty of the input hydrographs at each of the upstream boundary conditions was considered by sampling from the 100 class hydrographs. By assuming normal flow, the overall downstream valley slope, $b_c$, was used as downstream boundary condition. This assumption was considered in view of the lack of hydrograph information at the downstream boundary, however water-level predictions at the most downstream cross-sections can be associated with larger uncertainties due to this assumption (Pappenberger *et al*., 2006). Besides $b_c$, the channel-roughness coefficient, assumed uniform along all the channel length, $n_c$, and the floodplain-roughness coefficient uniform along all the floodplain, $n_f$, were also

considered to be uncertain parameters. Ideally roughness coefficients would be allowed to vary spatially to reflect changes in channel and floodplain characteristics (*e.g.* one value per reach or per each side of the floodplain), but this would have led to an increased number of parameters in the hydraulic model when there was not enough information at each reach to

constrain the local roughness.

Using a One-At-a-Time (OAT) design for sensitivity analysis, the effect that uncertainty in the channel depth and channel width (through a multiplying factor) had on the outputs was explored, which led to the incorporation of the channel-width multiplier ($w_f$) in the uncertainty analysis. For the hydraulic simulations, a total of 130 000 parameter sets were

sampled from a uniform distribution with ranges considered large but possible in the literature (Table 3) in the same way as for the RRM.

### 3.5.1  Output evaluation

Different degrees of belief ($d_i$) (Fig. 4) were obtained by comparing the simulations with the following evaluation data:

– One degree of belief value, $d_1$, as performance in predicting the maximum peak discharge value of point 3 (Figs. 1, 2 and Table 1).

– Two degrees of belief values, $d_{2-3}$, as performance in predicting time of the maximum peak discharge of points 3 and 6 (Figs. 1, 2 and Table 1).

– Ninety-nine degrees of belief values, $d_{4-102}$, as performance in predicting maximum

water levels along the main river and two tributaries (Fig. 2).

The fuzzy set values of *a* and *b* for evaluating the simulated peak discharge were set to 20% and 50% of observed value respectively. Thus, for differences between observed and predicted peak discharge within 20% of the observation, the degree of belief was assumed to

be equal to one and decreased to zero for differences larger than 50%. These values were chosen taking into account those values suggested by Benson and Dalrymple (1967) and (Jarrett, 1987). The fuzzy set values of *a* and *b* for evaluating the time of the peak were set equal to 0.5 and 2.5 hours respectively, thus the degree of belief for differences between observed and predicted time of the peak smaller than 0.5 hours was assumed to be equal to

one and it decreased to zero to allow for up to 2.5 hours of difference, an error considered possible in the observations. And finally, the fuzzy set values of *a* and *b* for evaluating the

water levels were set equal to 0.5 and 1.8 metres respectively. Thus, 0.5 metres was chosen to account for error in topography representation (Neal *et al.*, 2009), and 1.8 m was chosen considering the magnitude of the observed water level and that two years after the event witnesses' memories might have been associated with large uncertainties.

A parameter set was considered behavioural if the degree of belief was larger than zero for each of the 102 evaluation points. For every parameter set, a global score (*GS*) was calculated based on a weighted average of the degrees of belief obtained for each evaluation criterion.

$$GS = \sum_{i=1}^{i=102} w_i d_i, \tag{3}$$

where $w_i$ are the weights associated to the degrees of belief correspondent to the observations.
The weight associated to the peak discharge and the two times of the peak data ($d_{1-3}$) were set equal to 0.1 each, thus 0.7 was the weight corresponding to the sum of the degrees of belief associated to all the observed maximum water levels ($d_{4-102}$). A larger aggregated weight was given to predict the observed water marks in comparison to the peak discharge and times of the peaks to reflect the larger number of observed water marks (99) and because
focus was on predicting flood extent. The weights could be changed according to the purpose of the study which might also result in different ensembles being behavioural for different purposes (Pappenberger *et al.*, 2007).

Subsequently, likelihood values were obtained by scaling the global scores by a constant *C*, so they will sum to unity over all behavioural sets (Beven, 2009). Finally, the behavioural
parameter sets were used to generate a fuzzy likelihood water-level profile and map of the maximum flood extension during the Mitch event as in Di Baldassarre *et al.* (2010).

## 4    Results

### 4.1    Consistency in the post-event measured data

From prior inspection of the data, it was found that information about the maximum peak discharge and time of the peak were consistent (*i.e.* in comparison to locations at the upstream reaches): discharge values and time of the peaks were larger and later at downstream locations after the confluences. A plot of the high-water marks showed sudden jumps at some observation points without any obvious physical explanation, but this is perhaps to be

expected given the origin of those observations (witness accounts from memory). Thus, we did not eliminate any of the observations but instead allowed an uncertainty range associated with all observation points.

## 4.2   Representative hydrographs for the upstream boundary condition

Behavioural hydrographs to use as the upstream boundary conditions of the hydraulic model were obtained for the sub-catchments of the Grande, Guacerique and Chiquito Rivers and, by using behavioural sets at Grande and Chiquito River sub-catchments, at Salada Creek and Las Lomas Creek sub-catchments (Fig. 5). The cumulative distribution function (CDF) of the predicted peak discharge and of the time of the peak of 2 000, 8 000 and 9 000 behavioural

simulation for sub-catchments of the Chiquito, Guacerique and Grande Rivers respectively did not change significantly by adding 500 behavioural simulations more. Thus a total of 3 000, 9 000 and 10 000 behavioural simulations, obtained from a total of 61 205, 60 237, 60 833 samples respectively, were considered enough to infer 100 class hydrographs for the Chiquito, Guacerique and Grande Rivers sub-catchments respectively. When comparing the

prior and posterior distribution of the rainfall-runoff model parameters, five out of eight parameters were sensitive, the rainfall multiplier ($R$), rate of depletion ($m$), time constant ($t_d$), the main channel roughness coefficient ($n_{\mathrm{cu}}$) and maximum soil infiltration rate ($i_{\max}$) (Fig. 7).

## 4.3   Flood-wave propagation

There were no simulations for which all degrees of belief were larger than zero. Criteria $d_{1-3}$ were fulfilled by 47 894 out of 130 000 total simulations, but some observed water marks (criteria $d_{4-102}$) were constantly and largely under- or over-predicted. To allow for special cases, *i.e.* larger error in the observations or in the hydraulic simulations, the constraints were relaxed by allowing 10% of observed water marks (10 out of 99 observations) to be outside

the fuzzy bounds, *i.e.* the degree of belief was allowed to be equal to zero. By relaxing the constraints a total of 6 357 parameter sets were found, the degrees of belief for those parameters varied between 0.001–1, 0.04–0.96, 0.29–0.79 and 0.46–0.75 (for $d_1, d_2, d_3$ and average of $d_{4-102}$ respectively), and the global score (*GS*) from 0.40–0.78.

Change in the posterior distributions of the parameters showed that the channel roughness

coefficient and floodplain roughness coefficient were more sensitive than the channel width

factor and the slope for the downstream boundary condition (Fig. 8). Changes in the posterior distribution of the peak and time of the peak showed that the model was unsurprisingly more sensitive to input-hydrographs from the larger sub-catchments than from small sub-catchments (Fig. 9). Flood-wave propagation of different input-hydrograph combinations led to prediction of two markedly different times of the peak at the floodplain resulting in under-(over-) prediction when the earliest (latest) peak of input hydrograph combinations prevailed (Fig. 10).

There were three observed high water marks in the Chiquito River reach that were constantly under-predicted and outside the uncertainty bounds of the observations (Fig. 11). The propagation from the water-level uncertainty to the flood extent was more evident in urban areas, where the flood extent varies more with changes in the water level due to the presence of structural features such as buildings (Fig. 12). From behavioural simulations, the 90% confidence interval for prediction of the discharge at the floodplain outlet was 2 708 to 4 619 $m^3s^{-1}$ encompassing the 3 880 $m^3s^{-1}$ value estimated in JICA (2002) (reference point 7 in Fig. 1 and Table 1). For reference point 4, at Chiquito River, the 90% confidence interval was 247 to 482 $m^3s^{-1}$ also encompassing the 436 $m^3s^{-1}$ value estimated in JICA (2002).

## 5   Discussion

A field campaign after a large flood event is a possibility to collect information useful for flood forecasting and subsequent contingency planning in places where hydrometric measurements are lacking because of non-existing or broken gauges.

Our study demonstrated that it was possible, in a data-scarce situation, to reproduce an extreme flood event that was within the bounds of the uncertainty in the evaluation data. Our results support those of Bonnifait *et al.* (2009) and Ciervo *et al.* (2015) about the possibility to reproduce an extreme flood event by a suitable combination of RRM and hydraulic modelling tools with only event-based rainfall data and post-event hydrometric data. Here we additionally incorporated the GLUE methodology to account for expert knowledge of uncertainties in model parameters, rainfall input and evaluation data. Thus, the combination of a RRM with a hydraulic modelling tools within an uncertainty framework as in Montanari *et al.* (2009) and Pappenberger *et al.* (2005a) proved to be useful also in the case with only post-event-estimated hydrometric data.

After considering the uncertainties and their interaction it was possible to identify behavioural parameter sets that were used to obtain a realistic probabilistic reproduction of the flood-water level (Fig. 11) and flood extension (Fig. 12). In comparison to the deterministic estimates made by JICA (2002) using different modelling tools, in this work it was possible to obtain predictive ranges of the water level that encompassed most of the observations. The flood extent here, associated with a likelihood at each flooded cell, generally extended beyond the extent of the JICA (2002) mapping.

The combination of TOPMODEL and MCT allowed us to estimate behavioural hydrographs for the Chiquito, Guacerique and Grande sub-catchments. The simulations could be constrained (Fig. 6) in spite of the wide uncertainties in the data and the simplified assumption of the MCT routing for ungauged basins applied here. The rainfall multiplier ($R$), rate of depletion ($m$), time constant ($t_d$), the main channel roughness coefficient ($n_{cu}$) and maximum soil infiltration rate ($i_{max}$) were more important in selecting the resulting hydrographs (Fig. 7), whereas horizontal transmissivity ($T_o$), land-use coefficient ($l_u$), flood-wave celerity ($v_c$) were less sensitive.

The rainfall multipliers were sensitive and the means of their posterior distributions varied across sub-catchments (0.93, 1.5 and 1.3 for Chiquito, Guacerique and Grande respectively) (Fig. 7), suggesting that the spatial average rainfall estimated from the two available gauges was overestimated at Chiquito and underestimated at Guacerique and Grande sub-catchments. The Guacerique and Grande sub-catchments are larger and have higher topographic elevation than the Chiquito sub-catchment. Underestimation of rainfall for these sub-catchments might be the results of lack of stations to represent the rainfall spatial pattern, highly variable in the area (Westerberg *et al*., 2010). Thus, a simplistic account of a spatial and time averaged rainfall multiplier as in Fuentes-Andino *et al*. (2017) was also useful here to account for bias estimation of the spatially-averaged rainfall. The posterior distribution of the rainfall multiplier at Chiquito and Guacerique sub-catchments clearly aggregated to different mean values. The sensitivity to the multiplier was different in the case of the Grande sub-catchment, which also showed a different posterior marginal distribution shape for the rate of depletion ($m$) and time constant ($t_d$) (Fig. 7).

Different shapes of posterior marginal parameter distributions at Grande River sub-catchment relative to Guacerique and Chiquito River sub-catchments could be caused by parameter adjustment to fit the observations or by different hydrological processes going on in the

different sub-catchments. The sudden release of water from the dam could also be a reason for these differences. The posterior marginal parameter distributions for the Grande River sub-catchment suggest that it has shallower effective soil depth (low $m$) and a faster channel response in the MCT routing (low $n_{cu}$) than the other two sub-catchments. Hydrographs from a total of five sub-catchments (Fig. 6) from the TOPMODEL and MCT combination were used as upstream boundary conditions for the hydraulic simulations.

Even if more detailed post-event observations of flood extent might do better than water levels in constraining the LISFLOOD-FP (Fewtrell *et al*., 2011; Horritt and Bates, 2002), the modelling tool predicted the observed high-water marks, peaks and times of peaks well. Behavioural simulations for which the degree of belief for the peak discharge, time of the peak and at least 90% of predicted high-water marks (89 out of 99 observations) were above zero were identified.

The channel and floodplain roughness coefficients were the most important parameters for the hydraulic model (Fig. 8). As roughness coefficients directly affect the estimation of discharge and water level, the impact of their uncertainty has been shown previously in other studies (Dimitriadis *et al*., 2016; Pappenberger *et al*., 2005b; Warmink and Booij, 2015; Wohl, 1998). Here, uncertainty is expected to be particularly large as these coefficients interacted with uncertain post-event estimated discharge and high-water marks and also because they were assumed to be spatially-aggregated due to data limitations. For example, a more localised calibration of such coefficients could have helped to tackle the problem of localised channel erosion during flood events common in the area (Guerrero *et al*., 2012). Given the assumed spatial representation of the roughness coefficients and the uncertainty they are associated with, they interacted with all other sources of uncertainty in a complex way that is difficult to separate. Such complex interactions are contained implicitly in the resulting ensemble of behavioural simulations (Beven, 2016).

The effect of the input hydrographs from Grande River and Guacerique River sub-catchments on the resulting outputs is evident in Fig. 9. Thus, as in Dimitriadis *et al*. (2016), here the roughness coefficients and input flow were the most important sources of uncertainties. Two peaks in the input rainfall (Fig. 3) led to two main large peaks in the hydrographs as input boundary conditions (Figures 6 and 9). The propagation of input hydrographs along the floodplain led to under- or over-prediction of the times of peak (Fig. 10). This suggests that the spatial pattern of rainfall was not well represented by the gauge average, as also suggested

by the posterior distribution of rainfall multipliers in the RRM. Since rainfall data played an important role in predicting the times of peak, investment to improve the rainfall measurement system, *e.g.* radar estimates or a denser rain-gauge network, should be prioritized in the study area, especially because these data are easier to collect relative to discharge in a high

magnitude event.

Some observed high-water marks were constantly largely under predicted in the estimates by JICA (2002) and outside the prediction bounds produced here, even when allowing for significant uncertainty in the evaluation data (Fig. 11). Inspection at the points that were constantly under-predicted showed that no man-made structure could have been the reason for

such disagreement. Thus the problem of predicting at those locations could be caused by the inability of the hydraulic modelling tool to simulate the system under extreme conditions where effects such as sharp river bends might have an important local effect on the flow. However, a previous experiment using the one-dimensional HEC-RAS model on the same river also agreed with the results obtained here, and no localised effect in the under-predicted

places was obtained. Another reason for the disagreement could be large errors in the post-event data.

In general, minor errors between prediction and observations in this work could be caused by a weak spatial representation of topography and roughness coefficient, *i.e.* special topographic details in a highly populated area with man-made structures that could not be captured by the

DEM. However those local features might not affect the general flood extent (Haile and Rientjes, 2005).

The peak discharge at point 3 (Figs. 1 and 2) was under-predicted by most of the simulations (Fig. 10). However, the high water-mark was over-estimated at that location (Fig. 11). Reasons for this could be due to an over-estimation of the post-event peak discharge, or due to

an under-estimation in the observed high-water mark, or due to the simplistic representation of the downstream boundary condition assumed.

A general under-prediction of the water level in the Chiquito River reach could be due to the low (perhaps under-estimated) post-event-estimated peak discharge, as in comparison to the Grande and Guacerique sub-catchments, most of the hydrograph simulations for the former

were rejected because the simulated peaks were larger than the evaluations (even considering

the uncertainty) (Fig. 6). This could also be the reason for a lower rate of behavioural sets for the Chiquito River sub-catchment when comparing with the other two.

A detailed inspection of model structure, model set-up and data at specific points where the modelling tools did not perform well even after considering possible uncertainties in the parameters, input and evaluation data, could reveal areas for improvement.

This study was set up to demonstrate the use of post-event data and a combination of suitable RRM and hydraulic modelling tools with uncertainty analysis to reproduce an extreme flood in a data-scarce area. The behavioural ensemble found here depends on the uncertainties coming from the model structure (Dimitriadis *et al*., 2016), quality of the data (Pappenberger *et al*., 2006), topographic resolution (Haile and Rientjes, 2005), and spatial-aggregation of the parameters (Beven, 1995). Considering the dependency with those sources of uncertainties and their interaction, the post-event data proved to be useful in reproducing the Hurricane Mitch flood event. High-water marks obtained from personal memories of an event are a good source of information. To decrease uncertainty of such information, Institutions in charge of disaster prevention should be prepared to carry such surveys soon after flood events when memory is fresh. In fact soon after extreme events it is also possible to collect that information by surveying the marks left by the flood (*e.g.* Neal *et al*., 2009). Post-event-estimated peak discharge, though it is known to be associated with large uncertainties (Benson and Dalrymple, 1967; Jarrett, 1987), were a valuable source of information in this work. A higher spatial availability of flood peak discharge and time of the peak estimates would greatly benefit this methodology as it will allow a better quality control of individual estimates, to leave some of the estimates out for validation, and to estimate more localised pattern of roughness coefficients.

The use of this methodology can be done within a Bayesian framework in which the posterior distribution of the parameters is updated when more events become available. Data from more events could further reduce the predictive uncertainties and help us to learn from the flow behaviour at some localised areas where the errors were large. Post-event estimates in the future could likely also come from social-media information which is becoming gradually more available (Fraternali *et al*., 2012; Triglav-Čekada and Radovan, 2013).

The flood-hazard map presented here can be used by the committee in charge of disasters contingency and management in the City of Tegucigalpa (CODEM-DC) as a complement to

the 5–, 10–, 25– and 50–years return period hazard produced in JICA (2002), the 50–flood hazard produced in Mastin (2002) and Mastin and Olsen (2002) for spatial planning and to prioritise investment. If real-time discharge measurements are available to calculate the initial saturation of a catchment, behavioural parameter sets updated from a range of events can be used for forecasting the flood extent as shown by Romanowicz and Beven (2003) and Montanari *et al*. (2009). In the absence of such measurements, a guess of the initial discharge may also work since it will not significantly affect the prediction for the intense period of the event. Furthermore, for that period, our methodology can give a better performance since calibration is done against discharge, time and water level at the peak. It is also tempting to consider this methodology for forecasting fed both by an improved rain-gauge network and water-level information coming from social media.

## 6    Conclusions

In this study we tested the possibility to reproduce an extreme flood disaster in a data-scarce area, the devastating flood in Tegucigalpa triggered by Hurricane Mitch in 1998. It was possible to realistically reproduce this large ungauged flood event by using post-event hydrometric data in combination with rainfall data and various modelling tools, demonstrating the value of post-event field campaigns to constrain the uncertainties in estimates of hydrometric data, model parameters and output. A methodology has been proposed where post-event-estimated data are used to drive and constrain a combination of rainfall-runoff and hydraulic modelling tools to reproduce floods within a GLUE uncertainty-analysis framework. Results of the flood extent proposed here were comparable to the deterministic mapping produced by JICA (2002) using different modelling tools. However here more information was embedded as likelihoods of inundation associated with each cell in the floodplain.

Combining the TOPMODEL with the MCT routing to reproduce hydrograph in catchments with rapidly varied flow, *e.g.* release from a dam, resulted in hydrographs that were within the uncertain bounds of the observations. The predictive capability of the TOPMODEL and MCT combination warrants further exploration with more detailed and less uncertain event data. The rate and bias in the rejection of the hydrographs due to over-estimation, indicated under-estimation of post-event estimated discharge at one location. The propagation of estimated hydrographs through the hydraulic LISFLOOD-FP 2D resulted in successful predictions of observed high-water marks, discharge peaks and times of peaks within the uncertainty bounds

for most of the evaluation variables. A few critical locations in the floodplain were identified where the model set-up could not reproduce the maximum water level. Locations of disagreement between simulations and evaluations, after considering all important sources of uncertainties can provide information useful to improve model structure or post-event data-

estimation methods. Results showed the importance that rainfall data have in simulating the times of peaks, thus results would be improved by a better spatial assessment of rainfall. Improvements of this methodology can be done by using it within a Bayesian framework of updating the parameters posterior distribution when more events become available. The methodology proposed here can be useful for planning, prioritise investments and for flood

forecasting.

## Appendix

## Appendix A: Description of the TOPMODEL rainfall-runoff modelling tool

The TOPMODEL scheme in Fuentes-Andino *et al*., (2017) used here assumes a grid-cell distributed catchment. For any $n^{th}$ cell, the precipitation infiltrates first through the root zone

storage, with capacity equal to the minimum value between a constant and the local initial deficit ($D_n$) in units of length (L). The rate of infiltration is the minimum between the precipitation rate at that time or a specified maximum rate ($i_{max}$), in units of length divided by time (LT$^{-1}$), where the excess rainfall is routed as surface runoff. Once the maximum capacity of the root zone storage is reached water is leaked towards the unsaturated zone

storage ($S_{2_n}$) which has a maximum capacity equal to $D_n$ minus the root zone storage capacity. Once this capacity is exceeded, excess is again routed to the outlet as surface runoff. A rate $q_{v_n} = S_{2_n}/(D_n \times t_d)$ (LT$^{-1}$) infiltrates from $S_{2_n}$ towards a lumped subsurface storage, where $t_d$ is a local residence factor in LT$^{-1}$ units. Thus the catchment unsaturated zone recharge volume is estimated as the sum of all vertical flows:

$$Q_v = \sum ac_n \times q_{v_n}, \tag{A1}$$

for $ac_n$ equal to area of the cell.

Following the TOPMODEL concept (Beven, 1997, 2012; Kirkby, 1997), the following assumptions are done: (a) the saturated zone is in equilibrium with a steady recharge rate from an upslope contributing area ($a_n$); (b) the effective hydraulic gradient is assumed to be equal

to the local surface slope ($\tan \beta_n$); (c) horizontal a transmissivity profile is described by an

exponential function: $q_n = T_o \tan \beta_n \, e^{-D_n/m}$ ($L^2 T^{-1}$), which takes the value $T_o$ when the cell is saturated and has a rate of decline controlled by the parameter $m$. Following these assumptions, the downslope subsurface flow rate along the stream channel are summed to obtain the baseflow compounded volume in the catchment ($Q_b$):

$$Q_b = \sum q_n = A e^{-\gamma} e^{-\bar{D}/m}, \tag{A2}$$

Where $\gamma$ is the average soil topografic index, $\gamma_n = \ln(a_n/(T_o \tan \beta_n))$, of all the cells within a catchment, and $\bar{D}$ is the catchment mean storage deficit.

Equation A2 can be inverted to obtain an initial estimation of $\bar{D}$ by assuming an initial baseflow, then an estimation of the local deficit ($D_n$) is done through equation A3.

$$D_n = \bar{D} + m[\gamma - \gamma_n]. \tag{A3}$$

Update of the catchment average storage deficit is done at each time step by subtracting the unsaturated zone recharge ($Q_{v_{t-1}}$) and adding the baseflow ($Q_{b_{t-1}}$) from the previous time step:

$$\bar{D}_t = \bar{D}_{t-1} + \frac{[Q_{b_{t-1}} - Q_{v_{t-1}}]}{A}. \tag{A4}$$

15 Excess rainfall and water excess after the unsaturated zone storage that has reached its maximum capacity are routed towards the outlet using the network width function concept (NWF) (Kirkby, 1976; Surkan, 1969) which takes into account the structure of the river network when estimating the travel time from the n[th] cell to the outlet following the direction of flow. An adaptation by Grimaldi *et al*. (2010) was used here which assumes a varying

20 hillslope velocity ($v_{h_j} = l_u * \sqrt{s_j}$) dependent on the slope of the cell following the direction of the flow ($s_j$) and the land use coefficient, $l_u$. And keeping a constant celerity (Beven *et al*. 1979, McDonnell and Beven 2014).

Thus, the time spent by a surface water particle to travel from the n[th] cell to the outlet is estimated:

$$\tau_n = \sum_{j=1}^{j=N} \left[ \frac{l_h}{v_{h_j}} \right] + \frac{L_c}{v_c}, \tag{A5}$$

where, following the same path that the flow takes, there are a total of $N$ cells with length $l_h$ from the n[th] cell at a hillslope towards the junction at the channel. And $L_c$ is the length from

the junction towards the catchment outlet. Thus the final hydrograph at the outlet cell is equal to the sequence of compound runoff volume from cells arriving at the same time (estimated by Eq. A5) plus the groundwater contribution (A2) at those times.

## Appendix B: Description of the Muskingum-Cunge-Todini (MCT) routing

The Muskingum-Cunge-Todini routing (MCT) (Todini, 2007) used in this work was carried out using guidelines at Tewolde and Smithers (2007) to overcome the lack of river cross-sectional data. Thus to propagate a flood wave in a reach of length $\Delta x$, the following procedure was followed:

an initial guess for the outflow at the $t + \Delta t$ step ($O_{t+\Delta t}$) in units $m^3 s^{-1}$ is made using Eq. B1 and assuming $O_t \approx I_t$ for initial time step:

$$\hat{O}_{t+\Delta t} = O_t + (I_{t+\Delta t} - I_t). \tag{B1}$$

The reference discharge for the times $\tau = t$ and $\tau = t + \Delta t$, ($Q_\tau$) is given in Eq. B2:

$$Q_\tau = \frac{I_\tau + O_\tau}{2},$$

(B2)

and the reference water level, $y_\tau$, hydraulic radius $R_\tau$, average cross-sectional area velocity $v_\tau$, celerity $c_\tau$ and cross-sectional area $A_\tau$ in units of length, length, velocity, velocity and area respectively are estimated using the Manning's equation and some empirical relationships as

in Tewolde and Smithers (2007) (equations B3 to B7):

$$y_\tau = \left(\frac{Q_\tau \times n}{0.508 \times P_\tau \sqrt{S}}\right)^{3/5}, \tag{B3}$$

where $P_\tau = 4.75\sqrt{Q_\tau}$ is the wetted perimeter estimated for stable river channels, $S$ is the reach slope and $n$ the Manning's roughness coefficient.

$$R_\tau = \frac{2y_\tau}{3},$$

(B4)

where, Eq. B4 assumes a wide parabolic channel.

$$v_\tau = \frac{1}{n}(R_\tau)^{2/3}\sqrt{S}\,, \tag{B5}$$

$$c_\tau = 1.4 \times v_\tau, \tag{B6}$$

where a coefficient equal to 1.4 was chosen as the average between a parabolic channel and wide rectangular channel (1.2 and 1.6 respectively).

$$A_\tau = R_\tau \times W_\tau, \tag{B7}$$

where $W_\tau$ is the top flow width, assumed to be approximately equal to the wetted perimeter $(P_\tau)$.

The specialisation factor for correction of the Courant and Reynolds number, $\beta_\tau$, after Todini (2007) is:

$$\beta_\tau = \frac{c_\tau A_\tau}{Q_\tau}, \tag{B8}$$

thus the corrected Courant number, $C^*{}_\tau$, is estimated as :

$$C^*{}_\tau = \frac{c_\tau}{\beta_\tau}\frac{\Delta t}{\Delta x}, \tag{B9}$$

and the corrected Reynolds number, $D^*{}_\tau$ as:

$$D^*{}_\tau = \frac{Q_\tau}{\beta_\tau W_\tau S c_\tau \Delta x}, \tag{B10}$$

which yields to the following MCT parameters:

$$C_1 = \frac{-1+C_t^*+D_t^*}{1+C_{t+\Delta t}^*+D_{t+\Delta t}^*}; C_2 = \frac{-1+C_t^*-D_t^*}{1+C_{t+\Delta t}^*+D_{t+\Delta t}^*}\frac{C_{t+\Delta t}^*}{C_t^*} \text{ and } C_3 = \frac{1-C_t^*+D_t^*}{1+C_{t+\Delta t}^*+D_{t+\Delta t}^*}\frac{C_{t+\Delta t}^*}{C_t^*}, \tag{B11}$$

and the outflow at a reach at time $t + \Delta t$ is estimated by equation (B12):

$$\hat{O}_{t+\Delta t} = C_1 I_{t+\Delta t} + C_2 I_t + C_3 O_t. \tag{B12}$$

All the estimations for the time $\tau = t + \Delta t$ are computed twice to eliminate the influence of the first guess $\hat{O}_{t+\Delta t}$ in eq. B1.

## Appendix C: The Kuiper statistic test

The Kuiper statistic (V) (Kuiper, 1960) is estimated as the sum of the maximum negative and maximum positive distances($D_-$ and $D_-$ respectively) between two cumulative distribution functions ($S_{N1}$ and $S_{N2}$):

$$V = D_- + D_+ = max[S_{N1} - S_{N2}] + max[S_{N2} - S_{N1}]. \qquad (C1)$$

The significance level ($p$) is estimated by the following equation:

$$p = 2\sum_{j=1}^{j=\infty}(4j^2\,\lambda^2 - 1)e^{-2j^2\,\lambda^2}, \qquad (C2)$$

where

$$\lambda = V\left(\sqrt{N_e} + 0.155 + \frac{0.24}{\sqrt{N_e}}\right), \qquad (C3)$$

where,

$$Ne = N_1 N_2 / N_1 + N_2 \qquad (C4)$$

For $N_1$ and $N_2$ equal to the number of data points for first and second distribution.

## Author contribution

The experiment was designed by D. Fuentes-Andino, K. Beven, S. Halldin, C−Y Xu and G. Di Baldassarre. D. Fuentes carried out the experiment and performed the simulations. D. Fuentes-Andino prepared the manuscript with contribution from all co-authors.

## Acknowledgements

This research was carried out within the Universidad Nacional Autónoma de Honduras (UNAH) through agreement number 75000511–01 and the CNDS research school, supported by the Swedish International Development Cooperation Agency (Sida) through their contract with the International Science Programme (ISP) at Uppsala University (contract number: 54100006). The computations were performed on resources provided by SNIC through

Uppsala Multidisciplinary Center for Advanced Computational Science (UPPMAX) under Project p2011010 and the High Performance Computing Center North (HPC2N) under Project SNIC 2015/1–448. Thanks to the School of Geographical Sciences at the University of Bristol for useful support regarding the LISFLOOD-FP model.

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

Table 1 Post–event estimated peak discharge and time of peaks.

| Location | Discharge ($m^3s^{-1}$) | Time of peak (day h:min) | Source | Reference number (Figures 1 and 2) |
|---|---|---|---|---|
| Chiquito River | 167 | 31 Oct 00:00 | (Smith *et al.*, 2002) | 1 |
| Grande River | 2 340 | 31 Oct 00:00–02:00 | (Smith *et al.*, 2002) | 2 |
| Choluteca River | 4 360 | 31 Oct 00:30 | (Smith *et al.*, 2002) | 3 |
| Chiquito River | 436 | – | (JICA, 2002) | 4 |
| Guacerique River | 1 177 | 30 Oct 23:00 | (JICA, 2002) | 5 |
| Choluteca River | – | 31 Oct 01:00 | (JICA, 2002) | 6 |
| Choluteca River | 3 880 | – | (JICA, 2002) | 7 |

Table 2 Sampling parameter ranges to run the rainfall-runoff model

| Parameter | Abbreviation | Unit | Sampling range |
| --- | --- | --- | --- |
| Rainfall multiplier | R | (–) | 0.4–2.0 |
| Rate of decline of transmissivity | m | (m) | 0.005–0.035 |
| Horizontal transmissivity | $T_o$ | ($m^2\ h^{-1}$) | 0.001–20 |
| Time constant | $t_d$ | ($m\ h^{-1}$) | 1–60 |
| Land-use coefficient | $l_u$ | ($m\ s^{-1}$) | 0.04–0.2 |
| Flood-wave celerity | $v_c$ | ($m\ s^{-1}$) | 1.0–3.5 |
| Maximum soil infiltration rate | $i_{max}$ | ($m\ h^{-1}$) | 0.005–0.03 |
| Main channel roughness coefficient | $n_{cu}$ | ($s\ m^{-1/3}$) | 0.001–0.08 |

Table 3 Sampling range of parameters to run the hydraulic model.

| Quantity | Parameter | Abbreviation | Unit | Sampling range |
|---|---|---|---|---|
| 1 | Channel width factor | $w_f$ | – | 0.5–2.0 |
| 1 | Slope for downstream boundary condition | $b_c$ | % | 0.005–0.03 |
| 1 | Channel roughness coefficient | $n_c$ | $s\ m^{-1/3}$ | 0.005–0.3 |
| 1 | Floodplain roughness coefficient | $n_f$ | $s\ m^{-1/3}$ | 0.005–0.3 |
| 5 | Hydrograph for the upstream boundary condition (100 class hydrographs) | – | units | 1–100 |

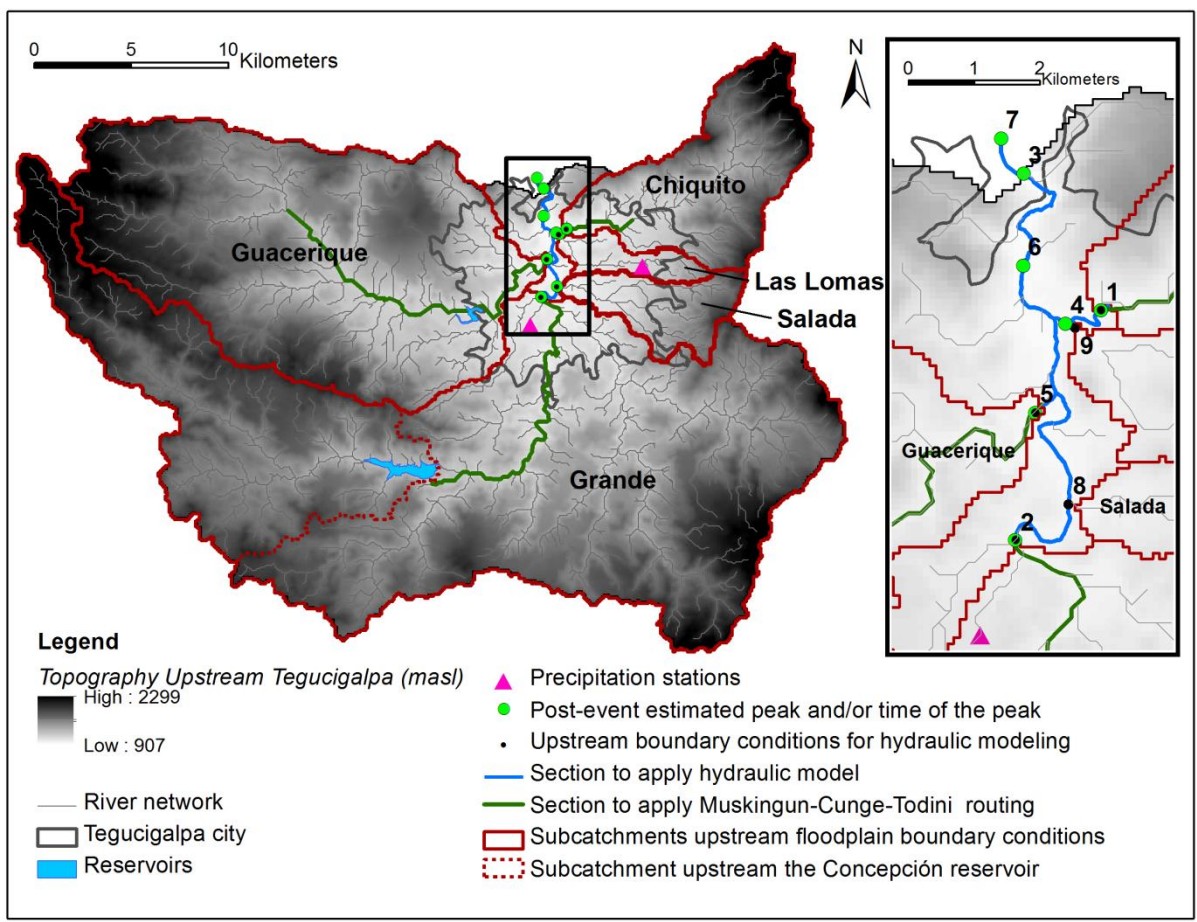

Figure 1 Study area and data location, Topography data from the Shuttle Radar Topography Mission (SRTM).

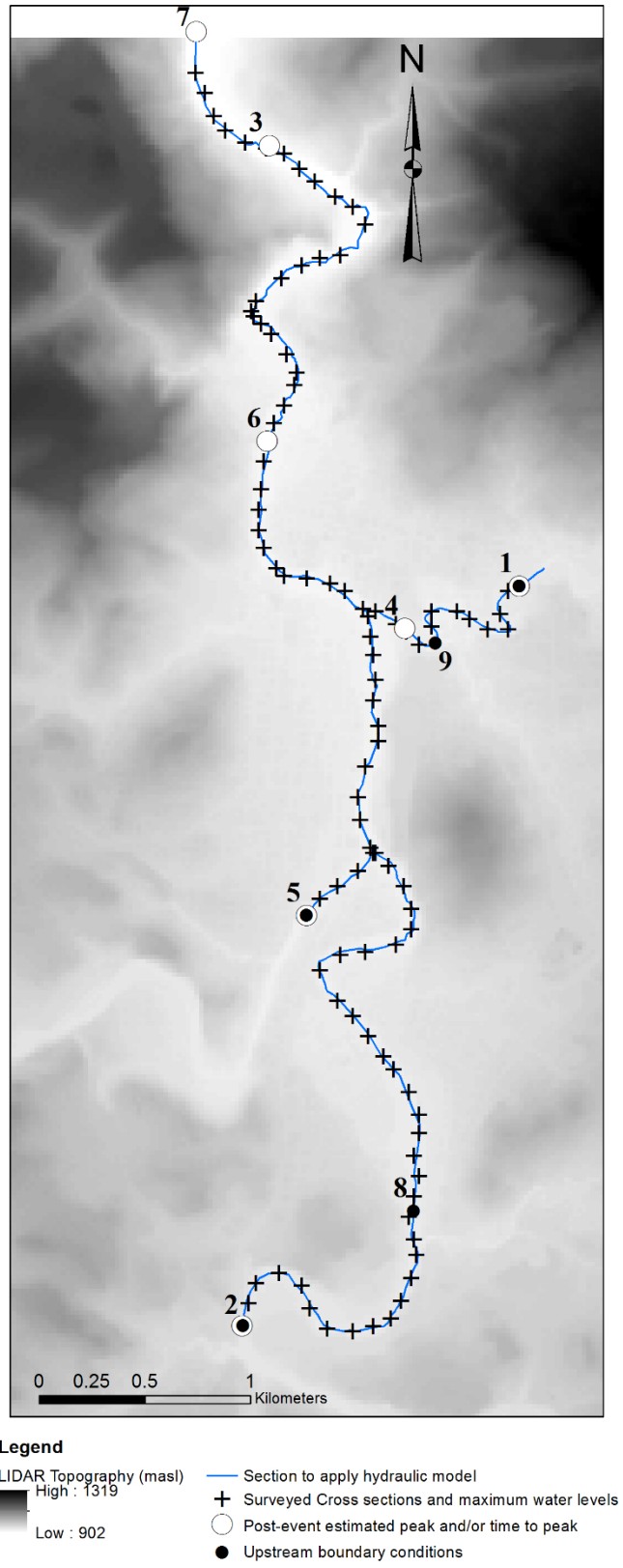

Figure 2 Geometry set-up for hydraulic simulation at the Tegucigalpa floodplain. Lidar data from Mastin (2002)

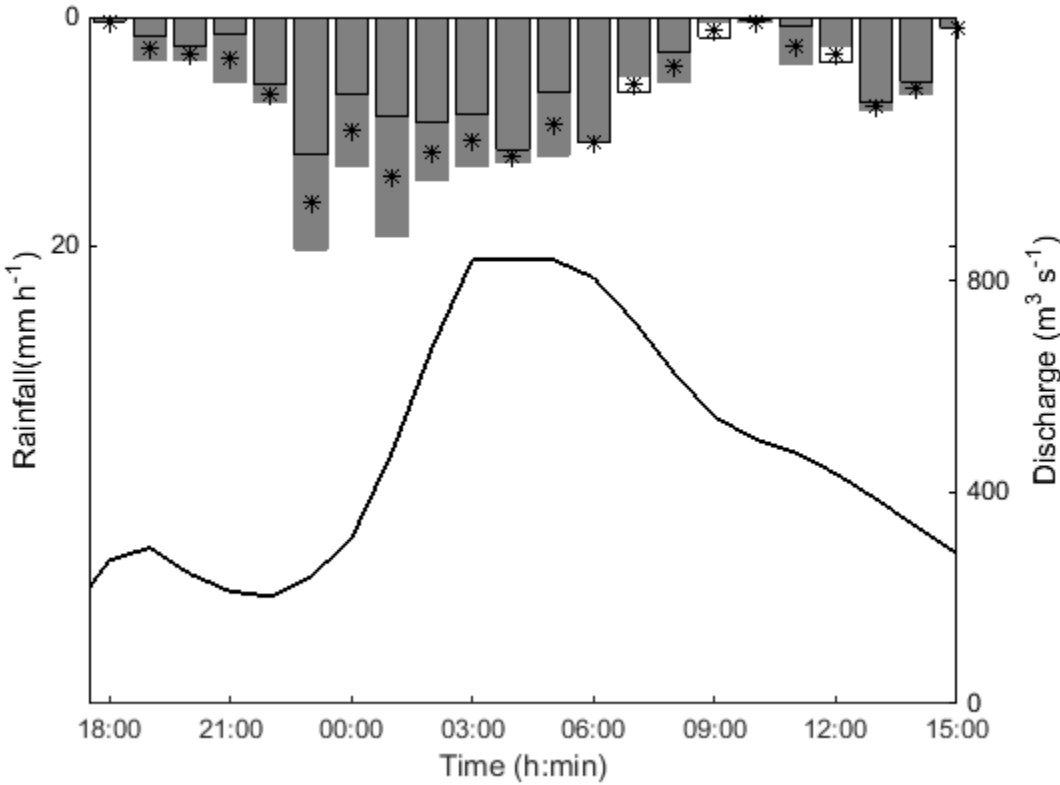

Figure 3 Hourly rainfall on 30–31 October 1998 at SMN station (grey bars), UNAH station (black outlined bars), average of the two stations (asterisks), and measured outflow at Concepción reservoir (continuous line).

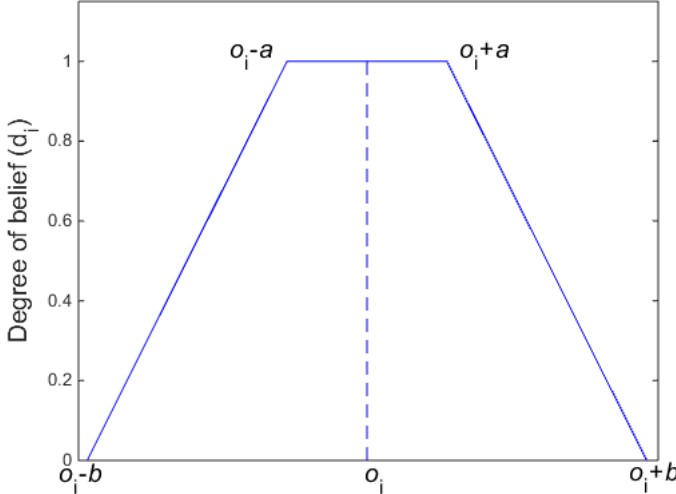

Figure 4 Fuzzy membership function for evaluation of model performance, $a$ and $b$ depend on the uncertainty associated with the evaluation ($o_i$).

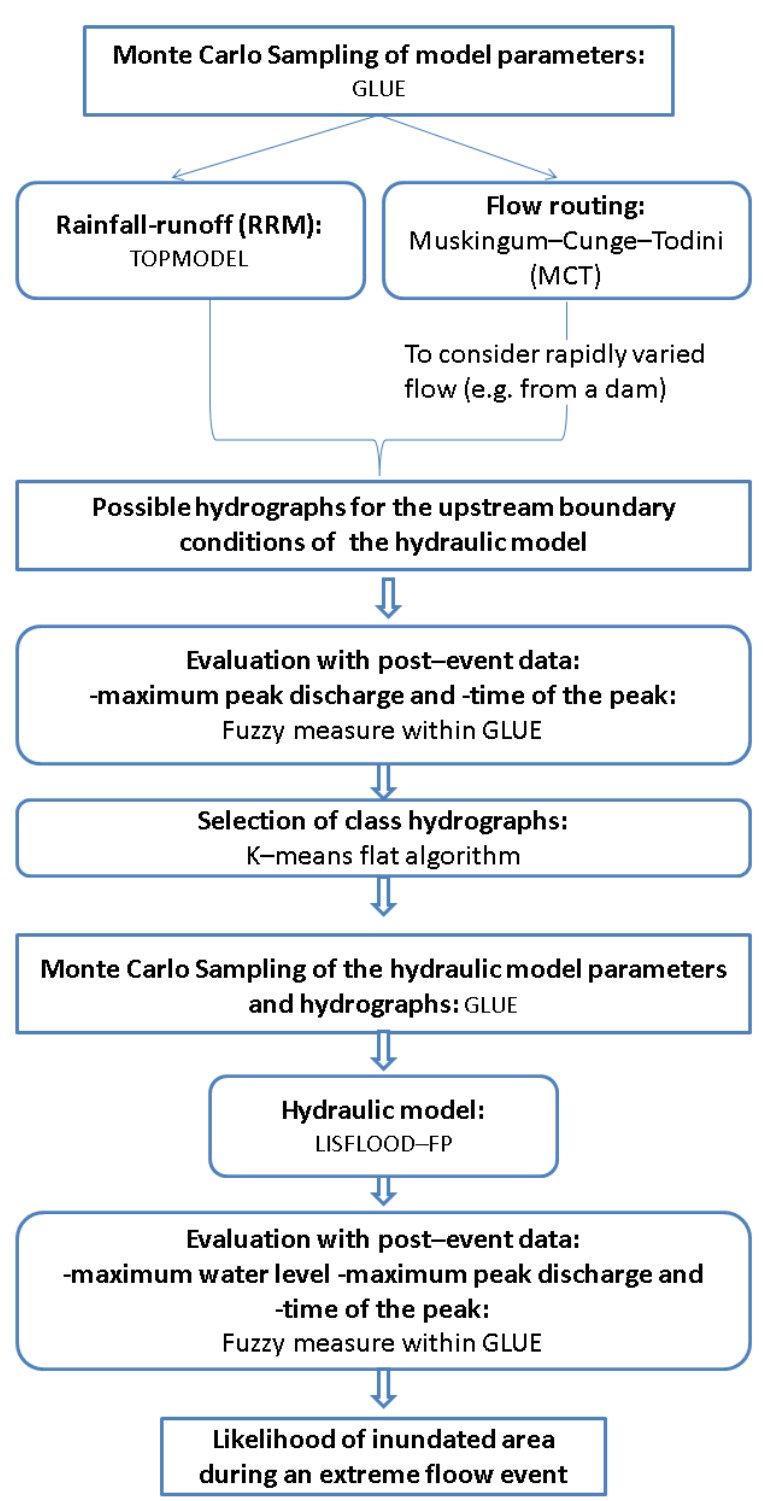

Figure 5 Scheme of the modelling framework used to reproduce an extreme flood event using post-event estimated data to drive and constrain a combination of modelling tools within an uncertainty analysis framework.

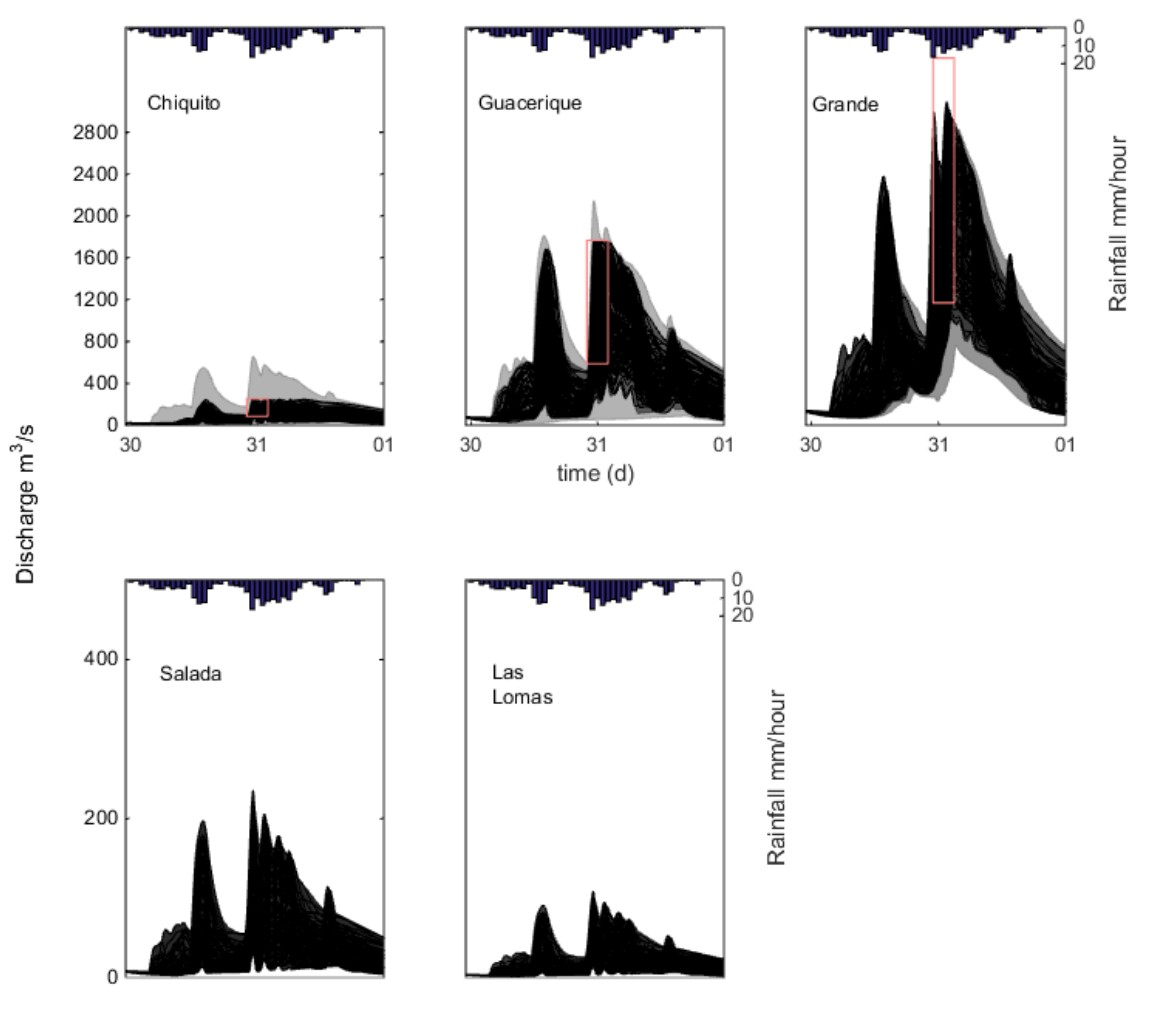

Figure 6 Precipitation (bars) and 100 class hydrographs chosen from the behavioural ones (black plots) for five sub-catchments upstream the floodplain. Predictive range of the 100% probability limits for all hydrographs simulations (grey shaded area) and rectangles representing the fuzzy set to allow for uncertainty for peak discharge and time of the peak for the sub-catchments of the Chiquito, Guacerique and Grande Rivers.

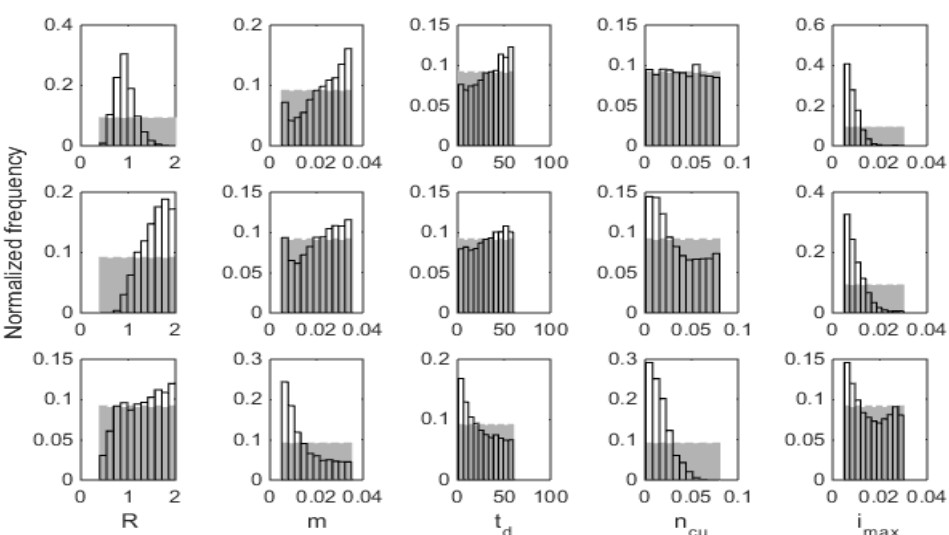

Figure 7 Prior (grey) and posterior (black outlined) relative frequency distribution for the for
the most sensitive Rainfall-Runoff parameters: rainfall multiplier ($R$), rate of depletion ($m$),
time factor ($t_d$) and the main channel roughness coefficient ($n_{cu}$) for the Chiquito,
Guacerique and Grande catchments (first, second and third row respectively).

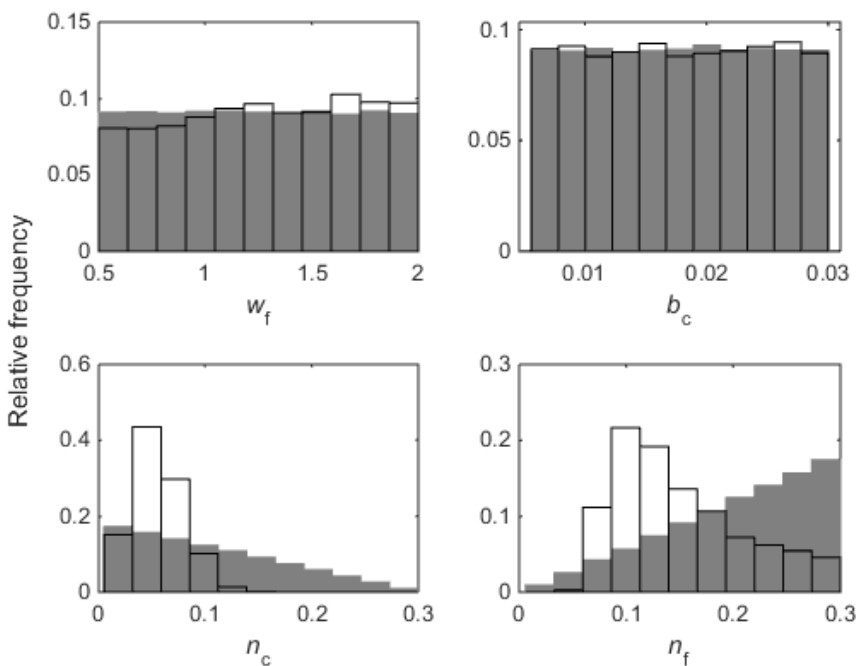

Figure 8 Prior and posterior relative frequency distribution (grey and black outlined bars respectively) of the LISFLOOD-FP parameters (width factor, slope for the downstream boundary condition, channel roughness coefficient and floodplain roughness coefficient, $w_f$, $b_c$, $n_c$ and $n_f$ respectively).

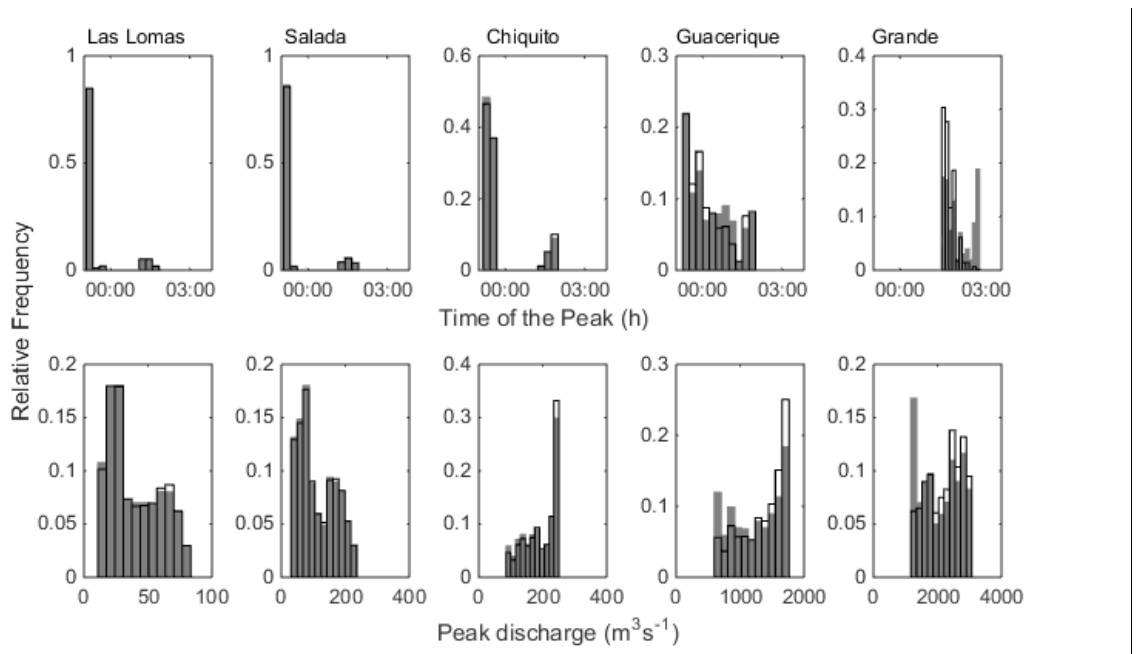

Figure 9 Prior and posterior relative frequency distribution (grey and black outlined bars respectively) of simulated maximum peak and time of the peak of input hydrographs for boundary conditions.

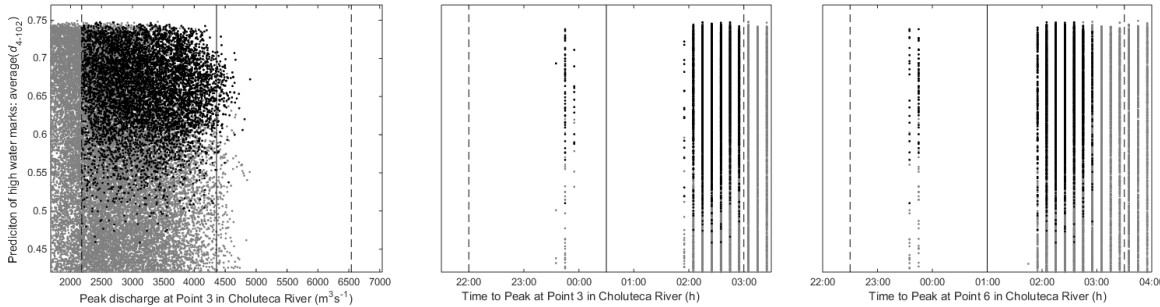

Figure 10 Performance of the model in predicting high-water marks, average ($d_{4-102}$), against predicted maximum peak discharge and two times of peak at Choluteca River (reference points 3 and 6 at Table 1) for non-behavioural simulations (grey dots), behavioural ones (black dots). Observed values and their limits of acceptability are plotted in continues and dotted vertical lines respectively.

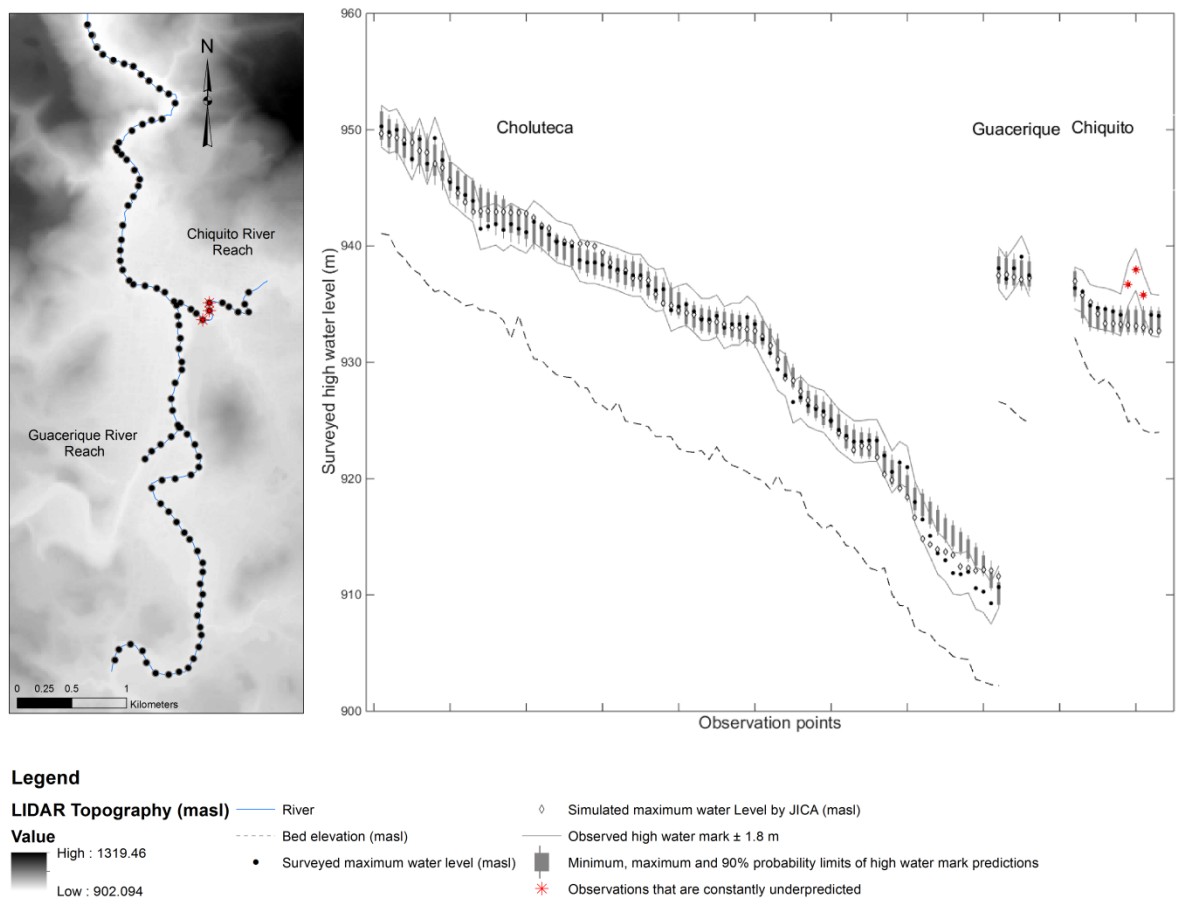

Figure 11 Likelihood of high-water marks during the Mitch event, considering uncertainty in model parameters, model input and evaluation data to drive and constrain a combination of rainfall-runoff and hydraulic modelling tools.

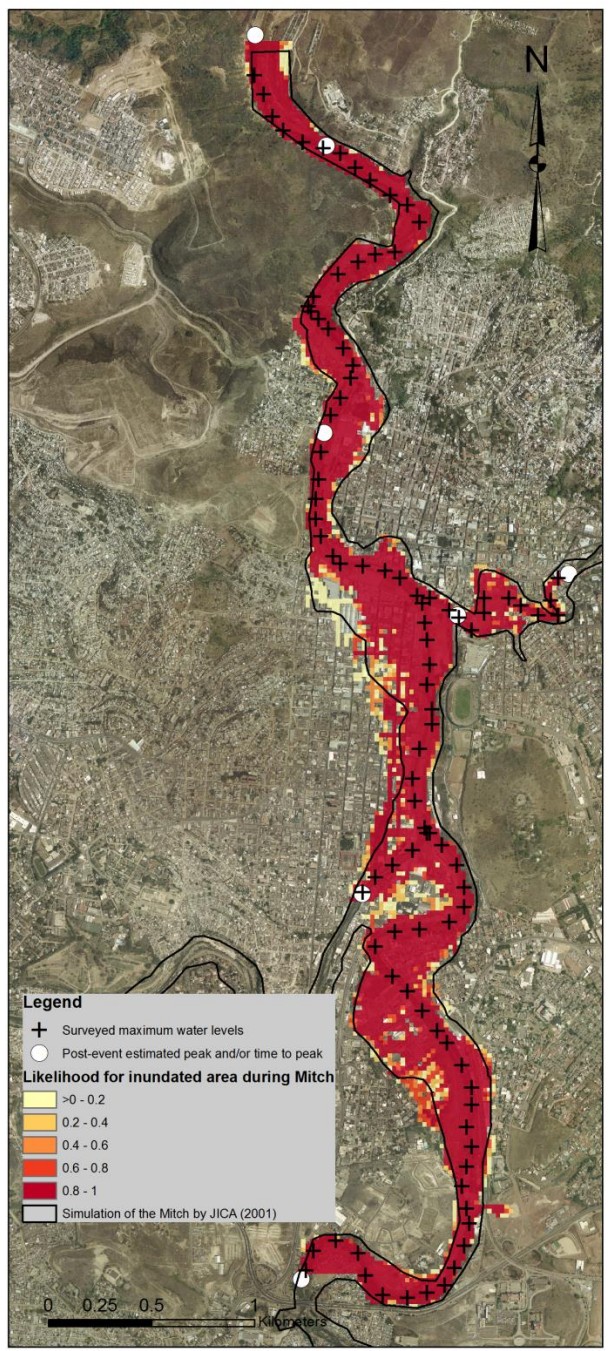

Figure 12 Likelihood of inundated area during the Mitch event on 30–31 October 1998,
considering uncertainty in model parameters, model input and evaluation data to drive and
constrain a combination of rainfall-runoff and hydraulic modelling tools. The deterministic
flood extent was obtained by digitalisation of the flood extend in JICA (2002).