# Peer review of "Reproducing an extreme flood with uncertain post-event information"

_Hydrology and Earth System Sciences, 2016_

## Short Comment (SC1) · 3 Oct 2016

This is an interesting work that shows a probabilistic inundation map originating from a combination of different models and sources of uncertainty. We propose the following and we think necessary improvements. In introduction several significant and similar works are missing from references concerning the development of a probabilistic inundation map framework (such as Apel et al. 2006, Aronica et al. 2002, Aronica et al. 2012, Di Baldassare et al. 2010, Horrit 2006, Merwade et al. 2008, Merz et al. 2008). In particular, we believe that the analysis of Aronica et al., (2002), Horrit (2006) is one of the very first analyses that introduce the concept and methodologies for probabilistic flood map rather than deterministic one. Indeed, it is rather impossible to deterministically account for all the various uncertainties affecting a flood inundation and therefore, a probabilistic concept must be introduced and applied for all flood

analyses. In pg. 10, ln. 5-12, the Authors estimate a minor effect from the errors in channel depth on the simulated water levels. Also, the Authors apply a uniform distribution on several sources of uncertainty, like the channel's and floodplain's roughness coefficients, the channel's width, the downstream valley slope, the input hydrographs etc. Indeed, the aforementioned parameters can be important factors of uncertainty as also verified in Dimitriadis et al. (2016), through the application of Monte Carlo techniques to benchmark tests. Particularly, they observe that from the applied sources of uncertainty, in the form of uniformly distributed hydrological and hydraulic parameters, important ones are the roughness coefficients in channel and floodplain, followed by the inflow discharge Q, the channel width (which equals the model resolution at the analysis) and the gradients of the channel and floodplain, with the latter corresponding to the channel depth and exhibiting the smallest effect to the overall uncertainty. Also, they observe that for approximately all tested models, numerical schemes and flow conditions, the uncertainty decreases with increasing discharge, longitudinal gradient and channel roughness coefficient, while it increases with increasing floodplain gradient , floodplain roughness coefficient and model resolution. These findings can be helpful in a real case study that is based on Monte-Carlo techniques, since they can be used by the modelers to limit down the parameters to the ones corresponding to higher uncertainties, and thus, to save valuable time.

References

Apel, H., Thieken, A. H., Merz, B., Blöschl, G., 2006. A probabilistic modelling system for assessing flood risks. Nat. Hazards 38, 79–100.

Aronica, G., Bates, P.D., Horritt, M.S., 2002. Assessing the uncertainty in distributed model predictions using observed binary pattern information within GLUE. Hydrological Processes 16, 2001–2016.

Aronica, G., Franza, F., Bates, P. D., Neal, J. C., 2012. Probabilistic evaluation of flood hazard in urban areas using Monte Carlo simulation. Hydrol. Process. 26, 3962–3972

Di Baldassarre, G., Schumann, G., Bates, P. D., Freer, J. E., Beven, K. J., 2010. Flood-plain mapping: a critical discussion of deterministic and probabilistic approaches. Hydrol. Sci. J. 55(3), 364–376

P. Dimitriadis, A. Tegos, A. Oikonomou, V. Pagana, A. Koukouvinos, N. Mamassis, D. Koutsoyiannis, and A. Efstratiadis, Comparative evaluation of 1D and quasi-2D hydraulic models based on benchmark and real-world applications for uncertainty assessment in flood mapping, Journal of Hydrology, 534, 478–492, doi:10.1016/j.jhydrol.2016.01.020, 2016.

Horritt, M. S., 2006. A methodology for the validation of uncertain flood inundation models. J. Hydrol. 326(1), 153–165

Merwade, V., Olivera, F., Arabi, M., Edleman, S., 2008. Uncertainty in flood inundation mapping: current issues and future directions. J. Hydrol. Eng. 13(7), 608–620.

Merz, B., Kreibich, H., Apel, H., 2008. Flood risk analysis: uncertainties and validation. Österreichische Wasser-und Abfallwirtschaft 60(5-6), 89–94

---

## Referee Comment (RC1) · Anonymous Referee #1 · 16 Oct 2016

The manuscript present an interesting work aiming at calibrating a series of rainfall-runoff and hydraulic models against data collected after the 1998 extreme flood occurred in the town of Tegucigalpa, the capital city of Honduras, flood induced by the Hurricane Mitch. Post-event surveys after large flood events have become more frequent over the years and are often based on the combination of field observations and hydrological and hydraulic modelling as a support for data interpretation (see Borga et al, 2008 for instance cited in the paper). The use of a Bayesian model calibration framework makes the originality of the presented manuscript. It can help to better assess to which extent post event survey data may or not help to constrain the values of the parameters of the models (i.e. may help to learn something about the behavior of the watersheds and the river network summarized in the values of the model parameters). If the idea of using a Bayesian calibration framework is interesting, its

implementation and the analyses of the results could be improved and deepened a lot. In a way, the impression is that the authors did only make part of the work : they did implement the framework and obtained results that are described but not really analyzed in the manuscript, but did not really put into question the procedure used, the associated score function and its influence on the obtained results. They do neither draw clear conclusions about what the event revealed or not about the hydrological or hydraulic processes. It seems, as illustrated by the title and the abstract that the authors are happy to have been able to reproduce the observed discharges and water level based on models. But this is not a surprise according to the large number of model parameters and the limited number of observed data and their inaccuracy. The problem was clearly over-parameterized which is illustrated by the remaining range of possible values for simulation results and parameter values (fig. 5 – 7). My feeling is that the methodology and its implementation did divert the authors from the real objective: analyzing post-flood data. This is why I did suggest major revisions for this manuscript that should provide both : critical analysis of the available post-event dada and more in-depth analysis of the proposed method and of the obtained results. Here are some questions raised by the presented results:

1) The post-event data set is extremely limited and uncertain. The stage-discharge relation is controlled locally by roughness coefficients. Uncertainties in discharge estimates will directly affect estimates of roughness coefficients and the inverse is also true. This indetermination can only be solved if other type of information is used: existing direct discharge measurements, flood wave propagation velocities (as suggested o p14 L 16). The available information is dense for the downstream 10 km river reaches. At least, a specific calibration of the hydraulic model could have been conducted using the information of points 2 and 8 or points 3, 6 and 7 (see figure 2), for a better assessment of hydraulic parameters.

2) All data cannot be considered as of equivalent value. Only 3 out of the 6 peak discharge estimates can be considered as based of field surveys. Discharge estimates 4

to 7 in table 1 are the result of the application of simple rainfall-runoff analysis. They should ideally not have been considered in the calibration procedure or with much larger uncertainties than the other indirect discharge estimates. Two additional estimates seem to have been available (peak outflow of Los Laureles dam and complete outflow hydrograph of the Conception reservoir. Why were these two values, probably relatively accurate if compared to other estimates, not used?

3) The consistency between the various post-event data should be checked either prior to their use for model calibration or even in the Bayesian procedure. This is true for peak discharge estimates upstream and downstream confluences (8,5,4,6) as well as for the high water marks a description of these marks (number, location) is missing in the manuscript).

4) The real efficiency of a global calibration procedure as proposed in the manuscript is questionable. The important features for the calibration may be different for each model (hydrological or hydraulic) and parameters. At least the weight given to each information in the score function should be put in question (a sensitivity analysis could be conducted).

5) Moreover, the number of calibrated parameters appears high and the correlation between these parameters may play an important role and reduce the capacity to narrow significantly the range for the posterior distributions. The number and range of possible values could be more limited for some parameters (again a sensitivity analysis could be useful). The correlation and dependence of the parameter values is an output of the Glue method: why is it not presented and commented? By the way, I wonder why the Glue procedure is still used in place of existing more elaborated Bayesian approaches : Bayesian MCMC (Gibbs algorithm for instance)...

6) The conclusions drawn in section 5 should remain prudent. The limited information about rain amounts and rates (2 gauges for a 800 km2 watershed) set a major limit to the whole study. Moreover there is clearly compensation between the introduced

rainfall multiplier and the parameters of the RR model that is complex to decipher (p 11, L 25).

7) The discussion part mentions some disagreements between model ouputs and observations that seem not to have been further analyzed: a) under-prediction of water levels (where? Are the water marks isolated, consistent with the surrounding marks or is there a specific problem for some reaches?), b) under-estimation of water levels at the Chiquito river but apparently over-estimation of discharges. The authors suggest some possible explanations P15, L5, but it is their duty to produce more than results and conjectures, but provide also explanations. Is the considered discharge estimate for the Chiquito river, probably the RR simulated discharge (see upstream comment), really accurate and consistent with the other estimate available upstream? A critical analysis of the data is unavoidable in such a study.

———————————————————

---

## Referee Comment (RC2) · Anonymous Referee #2 · 30 Oct 2016

The paper under consideration pertains to the extremely important issue of modeling floods in cases where the data is scarce or not available at all. Authors' ideas are illustrated by the flood event in Tegucigalpa triggered by Hurricane Mitch. Authors use the post event data collected two and even three years after the event. They propose an interesting modeling framework and discussion of the results. The nature of discussion obviously raises a lot of questions. One can easily think of a variety of other models that could be used and might lead to different conclusions. The material to verify the results is simply not solid enough but this is what the paper is about.

I read the paper with great interest as it touches upon the problem that hydrologists often have to deal with but to be honest, I am quite disappointed with the way the material was presented. It is not easy to follow the reasoning of the authors, not mention that one has to read this paper alongside with other publications to have the clear picture of

the subject. Please find below some remarks that in my opinion could make the paper more useful to the readers:

1. Authors do not mention various important sources of uncertainty such as insufficient knowledge of the analysed phenomena as well as the errors introduced by the models used in the calculations, which always simplify the described processes. Authors treat the models as black boxes, not even mentioning what assumptions and principles they use. Although TOPMODEL, MCT routing and LISFLOOD-FP are relatively well-known, in the paper I would prefer to have some basic information on the equations used, their dimensionality, model parameters etc. After all they present debatable simplifications and there exists a bunch of other models using different sets of equations, often treated as better representations of reality. One crucial issue in this respect is the way how hydraulic roughness is introduced in the model, since it creates one of the most important sources of uncertainty. Please refer to the in-depth discussion of such issues in Warmink and Booij (2015), Uncertainty analysis in river modelling, Rivers-Physical, Fluvial and Environmental Processes, GeoPlanet: Earth and Planetary Sciences, Springer, 255-277

2. Authors stick to the GLUE method as the framework for the uncertainty analysis without even mentioning that it is a quickly developing area and other methods could be successfully used in this context, to mention Markov Chain Monte Carlo algorithms, sequential Monte Carlo samplers, or likelihood-free algorithms such as Approximate Bayesian computations (ABC) as the implementation methods. A short discussion on the choice of GLUE method could be important in this respect.

3. Authors have a tendency to make the reader look for the information that could be easily provided in the paper. Some examples from pages 8 and 9:

When mentioning Kuiper statistic test, it would be useful to mention why this test was chosen. What makes it better than other much more popular statistic tests? Are we dealing with cycling variations?

"A stopping criteria as in Pappenberger et al. ( 2005b)" – do you mean one criterion or a few? Please provide or describe this criterion – the idea behind it.

What is K-means flat algorithm? Maybe an explanation that it refers to K-nearest neighbor classifiers (if it is the case) could give some more information to more advanced readers

In this point I am trying to convince authors to make the paper more self-contained, otherwise only the readers familiar with most of the tools used in the paper will benefit from reading it.

4. In the discussion part authors do not explain sufficiently why various disagreements between the model and observational results occur. I believe that a lot of them can be attributed to weak representations of the topography and roughness, which in such a complex catchment most likely cannot be represented by one parameter.

My overall opinion is that the paper is absolutely worth publishing, but the authors should attend to the above remarks in their revised manuscript. I am sure the paper will find a lot of readers, because it introduces the original methodology, discusses an interesting flood case study and obviously could be used to other data-limited events.

---

## Short Comment (SC2) · 31 Oct 2016

"Note to the editor and authors: As part of an introductory course to the Master programme Earth & Environment at Wageningen University, students get the assignment to review a scientific paper. Since several years, students have been reviewing papers that are in open online discussion for HESS, and they have been asked to submit their reports to the discussion in order to help the review process. While these reports are written as official reviews, they were not requested for by the editor, and we leave it up to the editor and authors to use these reports to their advantage. While several students were asked to review the same paper, this was not done to provide the authors with much extra work. We hope that these reports will positively contribute to the scientific discussion and to the quality of papers published in HESS. This report was supervised by dr. Ryan Teuling."

[Figure]

This paper answers the question to whether it is possible to achieve simulations of extreme floods, with limited data availability and large data uncertainties and can this simulation be truly useful for contingency planning and prevention. To answer this question information on discharge, extent of inundation and water level dynamics are required. However, hydrometric measurements of discharge and water levels during an event are often lacking or highly inaccurate, for example during the flooding caused by hurricane Mitch in Tegucigalpa, the capital city of Honduras. Instead of hydrometric measurements, information about water levels and discharges are inferred from post-event surveys.

In this study, post-event data have been used to calibrate hydraulic models. The GLUE framework has been used to account for uncertainty in hydraulic models and for the coupling of a Rainfall Runoff Model (TOPMODEL), with a hydraulic model (LISFLOOD-FP), using during-event measured data. Comparison of simulations and evaluations of these simulations were done by using a membership function of a fuzzy set to obtain a grade of degree of belief for each parameter set. Behavioural parameter sets were those for which all evaluation variables fell within the support of a fuzzy set defined by the uncertainty range associated with the post-event estimated evaluation data. These behavioural parameter sets were used to generate a fuzzy likelihood water level profile and map of the maximum flood extension during the Mitch event. The paper concludes that it is possible, considering the uncertainty in post-event data, to reasonable reproduce the extreme Mitch flood in spite of no hydrometric gauging during the event.

The paper fits well in the scope of the journal by addressing a new interesting modelling approach to reproduce floods with post-event data. The paper gives a clear overview of the already existing methods and approach in literature. The structure of the method is well thought through by first introducing the modelling framework where after the used models are discussed. However the paper is very interesting, some improvements should be made concerning the relevance, the added value of the used modelling framework and the validation of the model. So, the idea of the paper is nice although

some improvements should be made, to make the paper more convincing.

Although, the introduction of the paper gives a broad picture of the already existing methods and models, the objective does not become clear from the introduction. The objective could be to reproduce the large ungauged flood event by using post data. However the objective could also be to reproduce the water level dynamics of extreme events in order to improve model structures, which corresponds to the main research question. The main research question was "To whether is it possible to achieve simulations that can be truly useful for contingency planning and prevention?" However this question is not answered in the conclusion. From the conclusion, the objective seems to be to reproduce the flood. If that is the case, the question arises: what can you do with this information? Horrit et al (2002), Papenberg et al. (2006) and Ciervo et al. (2015) suggest using the flood inundation area to improve flood forecasting systems. So maybe this can also be the case in Honduras. Is there a flood forecasting system in Honduras which can benefit from this knowledge or can such a system be developed? Some additional information about the applications of the produced inundation model should strengthen the social relevance of the paper.

In the discussion, the authors give a good explanation of what is done and what are the results of the TOPMODEL and the LISFLOOD-FP. Also, they compare the results with the results of Bonnifait et al. (2009). Bonnifait et al. (2009) use a combination of TOPMODEL and LISFLOOD-FP but without the GLUE network. They found discharges $\pm$ 10%, while the paper found discharges in a range between 2708 and 4619 ms-1 with a 90% confidence interval. In my opinion the results of Bonnifait et al. (2009) and Fuentes-Andino are more or less the same. So the question arises what is the added value of the GLUE framework to the modelling approach of combining a RRM with a hydraulic model. Since both approaches give more or less the same result. Smith et al. (2002) did also research about the peak discharges caused by Mitch. Using standard USCG techniques, he found discharge value within the range proposed by Fuentes-Andino for the same reaches. So why using model techniques while standard USCG

techniques give more or less the same result concerning the discharge.

A third weakness of the model development is the lack of validation of the model. All the model runs with different parameters are evaluated against a fuzzy membership. However the valid parameter sets are not validated against other data. Since discharge is only evaluated against one point and time of peak against two points in the river, validation is necessary. Papenberg et al. (2006) suggest that the prediction of current flood models is the subject to errors in input data, model structure and the observations used in model evaluation. This statement proves the importance of validation. Probably the authors have decreased these errors due to the GLUE framework. However it would be nice if they prove this by doing some validation. Validation can be easily done against pictures (Papenberg et al. (2006) or remote sense maps (Horrit et al. (2002). Another possibility is to calibrate the model against discharge measurements and validate the same model to its prediction of flood extent. A good model should be give accurate prediction of both, discharge and flood extent (Horrit et al. 2002). Another possibility is to calibrate the model against regular runoff events and validate it against the extreme flooding as is done by Grillaskis et al (2010). Adding validation to the proposed method can prove the added value of the approach.

Furthermore there are some minor issues which need some more explanation. In the article is often referred to the JICA report (2002) but this report is nog public assessable. Since quit a lot of data are used from this report, more explanation of the findings from the report will be desirable. For example, peak discharges at different locations were estimated by JICA (2002) and also the maximum water levels were surveyed post-event by JICA (2002). However, the exact approach to obtain these data remains unclear.

Page 1, 4 and 25: In the abstract, the area description and the caption of Figure 1 Honduras is not mentioned. It would be nice if this can be added to these sections. Most people are not familiar with Tegucigalpa.

Page 6, line 13: The statement "Propagation of the water level uncertainty in the flood extent was more evident at highly dense urban areas" is made. However, it does not become clear how you can conclude this from a likelihood map for inundation.

Page 8 lines 9-13: In this section six model parameters are mentioned. However it is more easily understandable if the model for which the parameters are used, is mentioned as well.

Page 22 table 1 gives estimations of time of peak, but not for all the points. Why is this the case? Furthermore points 8 and 9 which are visible in figure 1 and 2 are not described in the table.

Page 25 and 26: The caption of figure 1 and 2 should be referred to the points in table 1 and 2. Without referencing in the caption it is not clear what the numbers indicate.

Page 25: Figure 1 shows the study area. However, the topography and the course of the river outside the study area are not shown. Expanding the figure a bit will give more knowledge about how the river continues outside the study area. Expanding the figure will also help to place the study area somewhere in Honduras.

Page 30, figure 6 shows only the sensitive parameters. Why does this figure not show all the parameters? Showing all the parameters will convince people with the result.

Page 34: Figure 10 is unclear. The symbols of the likelihood of high-water-marks are too small and cannot be distinguished from each other. This figure should definitely be improved to be able to draw conclusions or observations from it.

References

Bonnifait, L., Delrieu, G., Le Lay, M., Boudevillain, B., Masson, A., Belleudy, P. & Saulnier, G. M. (2009). Distributed hydrologic and hydraulic modelling with radar rainfall input: Reconstruction of the 8–9 September 2002 catastrophic flood event in the Gard region, France. Advances in water resources, 32(7), 1077-1089.

Ciervo, F., Papa, M. N., Medina, V., & Bateman, A. (2015). Simulation of flash floods in ungauged basins using post‐event surveys and numerical modelling. Journal of Flood Risk Management, 8(4), 343-355.

Grillakis, M. G., Tsanis, I. K., & Koutroulis, A. G. (2010). Application of the HBV hydrological model in a flash flood case in Slovenia. Natural Hazards and Earth System Sciences, 10(12), 2713-2725.

Horritt, M. S., & Bates, P. D. (2002). Evaluation of 1D and 2D numerical models for predicting river flood inundation. Journal of hydrology, 268(1), 87-99.

JICA (2002) On flood control and landslide prevention in Tegucigalpa metropolitan area of the republic Honduras, Pacific consultants International and Nikken consultants, Tegucigalpa, Honduras

Pappenberger, F., Matgen, P., Beven, K. J., Henry, J. B., Pfister, L., & Fraipont, P. (2006). Influence of uncertain boundary conditions and model structure on flood inundation predictions. Advances in Water Resources,29(10), 1430-1449.

Smith, M. E., Phillips, J. V., & Spahr, N. E. (2002). Hurricane Mitch: Peak discharge for selected river reaches in Honduras. US Geological Survey, US Department of the Interior.

---

## Short Comment (SC3) · 2 Nov 2016

Note to the editor and authors: As part of an introductory course to the Master pro-gramme Earth Environment at Wageningen University, students get the assignment to review a scientific paper. Since several years, students have been reviewing papers that are in open online discussion for HESS, and they have been asked to submit their reports to the discussion in order to help the review process. While these reports are written as official reviews, they were not requested for by the editor, and we leave it up to the editor and authors to use these reports to their advantage. While several students were asked to review the same paper, this was not done to provide the authors with much extra work. We hope that these reports will positively contribute to the scientific discussion and to the quality of papers published in HESS. This report was supervised by dr. Ryan Teuling.

[Figure]

General comments:

This study investigated if it is possible to achieve simulations that can be truly useful for contingency planning and prevention when limited data is available with large uncertainties. In order to answer this, an investigation which combines RRMs, hydraulic models and post-event data within an uncertainty analysis framework has been done for the extreme flood-event of 1998 in Tegucigalpa, Honduras. The models that have been used, TOPMODEL, MCT-routing, LISFLOOD-FP hydraulic model and GLUE, simulate the runoff and river flows very good within a very good uncertainty-analysis framework. For this investigation high water marks of the event in 1998 were gathered via interviewing residents who experienced the event at specific cross-sections in the area. The parameters within the hydraulic model are based on the degree of belief at each observation point. These parameters were used to generate a fuzzy likelihood water level profile and map the maximum flood extension during the Mitch event. This investigation could be very interesting for flood areas where limited data is available. Because the uncertainties of the different models, parameters and the data itself have been included. Therefore the results should be more realistic and more useful for contingency planning and prevention. However, the model evaluation shows its shortages and gives therefore not the wanted results. Therefore I suggest major revisions for this manuscript that should incorporate : the formulation of an alternative hypothesis and a critical analysis of the model and the uncertainties that are used in this investigation. My suggestions for this paper are explained further in the text.

Major arguments:

Hypothesis

The aim of this investigation is to know if it is possible to achieve simulations that can be truly useful for contingency planning and prevention when limited data is available with large uncertainties (P2, L25; P3, L6). The hypothesis that is stated in the paper gives only a part of a bigger investigation, analyzing extreme floods to prevent such
extreme events, which is more interesting and needed. Therefore a new hypothesis should be formulated or added. Suggestions to reformulate the hypothesis can be found in the paper by Beven (2001). He formulated an hypothesis: 'The fast response of a catchment is dominated by surface runoff derived from rainfall' which can be used to compare different models which have required components and parameters for hydrograph simulation. Another suggestion for a new hypothesis is : 'This 'new' flood inundation method which incorporate uncertainty in its model and (post-event) data give more reliable results than other flood inundation methods'.

Model evaluation

In order to answer the hypothesis, an investigation which combines RRMs, hydraulic models and post-event data within an uncertainty analysis framework has been done (P3, L29 to P4, L3). Other, more easily, investigations which could also give an flood-inundation map are however not given. The model has not been compared with other models, while this is very important (Beven 2001). Some suggestions (many other approaches are possible, I just named two new approaches):

1) test if the same results would occur if uncertainty has been added to the results of JICA (P12, L14-18)

2) asses the spatial likelihood of flooding hazard using naïve Bayes and GIS (Liu et al. 2015)

3) asses the uncertainty in distributed model predictions using observed binary pattern information within GLUE (Aronica et al. 2001-2016)

A main subject in this investigation is the uncertainty of the different models, parameters and the data itself (P1, L21-26). A logic reasoning would be to test the combination of the different models that has been used on real existing data, which is present at the moment (Mastin, 2002). Only then the uncertainty of the model is tested in a trustworthy way for normal and extreme events. Therefore I would like to see the test-results

when the model has been tested with existing data to proof that the model outputs give reliable results with less data. Overall, I would recommend to evaluate the model with real existing data and to test if the approach that is described by the paper is better than other approaches. If so, the paper can contribute to the prevention of floods in a large part of the world.

Uncertainty

As I already stated the uncertainty of the different models, parameters and the data itself is a very important subject in this investigation(P1, L21-26). However these uncertainties in the parameters and data have not explicitly been analyzed in the manuscript. Some suggestions:

1) The comparability between the different post-event data should be checked. This holds for the peak discharge estimates and for the high water marks. Peak discharges at points 1 and 2 were estimated using the width contraction analyses, at point 3 the discharge was estimated by the slope-area analyses and the discharges at points 3, 5 and 7 were estimated by a rainfall-runoff analysis (P5,L25 to P6,L5). These discharges cannot be seen as equivalent values and need therefore different and (larger) uncertainties. The high water marks data should be analysed on their comparability with each other and by their corresponding discharge. The description of high water marks (location and number) should also be added in the manuscript (annexes). A calibration with real existing data is therefore needed as described above.

2) There are a lot of parameters used in this analysis which have a large number and range of possible values. This could be limited by, for example, a sensitivity analysis (Griensven, 2006). The correlations between the different parameters could also be described more clearly. A specific point that can be made here, is the water level – discharge relation which is locally controlled by roughness coefficients (P8, L12-16 ; P10, L8-9). Uncertainties in both, can influence each other. Smaller uncertainties can be obtained through specific calibration with existing data, as mentioned before.

Minor arguments:

The assumed uncertainty range for the peak discharge, assuming all predications within the fuzzy set equally good was $\pm$ 50

The fuzzy set values of a and b for evaluating the simulated peak, time of the peak and water levels were set to 20

The values of the degree of beliefs of water level data varies between 0.46 and 0.75 with relaxed criteria are quite acceptable (P14, L6-8). I would like to know why this is acceptable and why this would give trustworthy results.

The TOPMODEL parameter sets from Grande and Chiquito River sub-catchments were used to simulate the hydrographs at Salada creek and Las Lomas creek sub-catchments (P11, L10-13). The reason for this is missing.

Minor issues:

Abstract : Tegucigalpa is the capital of Honduras, is missing.

Methods : The combination of the different models (TOPMODEL, MCT-routing, LISFLOOD-FP hydraulic model) and the models independently, within the GLUE uncertainty analysis framework, are not clearly explained for a broader public. This could maybe be done by a sketch of the combination of the different models.

P10, L19-24 : Reference is done to Table 1. In my opinion the main point of this reference is to show where points 3 or 6 are positioned. This could be better done if the reference is to Figure 2.

P11, L19-20 + Fig 6: 'five out of eight parameters were sensitive (Fig 6)' It would be better to show the eight parameters. The sensitivity of the different parameters can then be observed.

P14, L18 : 'rainfall data played an important role' is not further explained or referenced.

P19, L7-9 : 'JICA: On flood control and landslide prevention in Tegucigalpa metropolitan area of the republic of Honduras, Pacific consultants International and Nikken consultants, Tegucigalpa, Honduras., 2002.' This report cannot been found on the internet.

P25, Fig 1 : Implement Tegucigalpa is capital of Honduras and show this on a bigger 'world' map.

P26, Fig 2 : Figure is not detailed enough. Show the different catchments as has been done in the right figure of Figure 1.

P27, Fig 3 : The figure is not very valuable for understanding the story. Therefore it is not needed.

P35, Fig 11 : The figure is very unambiguously. If there is a likelihood of inundations than this likelihood is mainly 0.8-1. This is very peculiar.

References:

Aronica, G., P. D. Bates, and M. S. Horritt. "Assessing the uncertainty in distributed model predictions using observed binary pattern information within GLUE." Hydrological Processes 16.10 (2002): 2001-2016.

Beven, Keith. "On hypothesis testing in hydrology." Hydrological processes15.9 (2001): 1655-1657.

Van Griensven, A., et al. "A global sensitivity analysis tool for the parameters of multi-variable catchment models." Journal of hydrology 324.1 (2006): 10-23.

Liu, Rui, et al. "Assessing spatial likelihood of flooding hazard using naïve Bayes and GIS: a case study in Bowen Basin, Australia." Stochastic Environmental Research and Risk Assessment (2015): 1-16.

Mastin, Mark C. Flood-hazard mapping in Honduras in response to Hurricane Mitch. US Department of the Interior, US Geological Survey, 2002.

---

## Short Comment (SC4) · 8 Nov 2016

Note to editor and authors: The course "Research Trends in Physical Geography and Hydrology" offered at the Department of Earth Science at Uppsala University, requires that each student select a paper and submit a review as part of the assessment of the course. The instructors also encourage that participants upload their reviews as part of contributing to a scientific discussion. This report is submited in that spirit and supervised by Prof. Giuliano Di Baldassarre.

General Comments:

The main objective of the paper is to reproduce an extreme flood event in a region in Central America where there is both paucity and data uncertainty. A combination of both post-event data regarding water levels and river discharge was used in calibrating;

[Figure]

Rainfall-Runoff Model (RRM), routing and a hydraulic model for the modelling task. And uncertainties were accounted for using the Generalized Likelihood Uncertainty Estimation (GLUE) Framework.

The structure of the methodology part of the work was presented clearly for the most part, and followed the order at which the task was carried out. Fuzzy set theory introduced by Lofti Zadeh (Zadeh, 1965) was used by the authors to represent uncertainty limits for post-event variables. And also used as a likelihood function to accept behavioral models. This is one theory that is obviously not new in hydrology (Bardossy et al., 1990) but still trailing the classical 'probabilistic methods' for quantifying uncertainty. In my opinion, there should be a short paragraph describing why it was chosen and adapted in this case, and provide references for the interested reader who wants to follow-up on the fuzzy theory.

Major Comments:

The choice of using fuzzy methods to characterize uncertainty made sense, since the objective of the paper is reproduction of a flood event that has occurred in the past, and not looking at the stochastic behavior of observed flood samples. This is clearly captured by Zadeh in his quote "the notion of fuzzy relates to a situation in which the source of imprecision is not a random variable or stochastic, but rather represents classes which do not possess sharp boundaries" (Zadeh, 1990). Following Zadeh's remark as a basis for reasoning, the uncertainty limit around the best estimate of the variable (post-peak discharge), can be represented as a range of possible values. My critique is based on the choice of the membership function.

One of the strong points of fuzzy set theory is that it employs non-complex shapes to characterize uncertainty, however these shapes are still amenable to explanation. The selection of the trapezoidal membership function for this study seems to apply to a phenomenon that, at first approximation, will result to an interval number. For instance, the capacity of a machine to lift a dead weight ranges from 230 - 350 Joules/seconds,

or that the kettle is "hot", were hot covers a range of values. In reconstructing a flood that happened in the past, the membership function that represents this case is the triangular membership function. This is because in estimating the post-peak discharge using either the rating curve or any other means, a single "best" estimate is arrived at as a first approximation, then the support of the membership function represents the uncertainty or classes of possible values as suggested by Zadeh.

In general the study and the methodology applied is quite useful for catchments with little or no data at all, the so-called "ungauged basins". The authors' recommendation of social media data is quite inspiring as a source of information to constrain model prediction uncertainty.

References

Bardossy, A., Borgadi, I., & Duckstein, L. (1990). Fuzzy Regression in hydrology. WATER RESOURCE RESEARCH, 26(7), 1497–1508.

Zadeh, L. A. (1965). Fuzzy sets. Information and Control, 8, 338–353.

Zadeh, L. A. (1990). Fuzzy Sets and Systems. International Journal of General Systems, 7, 129 –308.

---

## Short Comment (SC5) · 11 Nov 2016

Note to the editor and authors: As part of an introductory course to the Master programme Earth & Environment at Wageningen University, students get the assignment to review a scientific paper. Since several years, students have been reviewing papers that are in open online discussion for HESS, and they have been asked to submit their reports to the discussion in order to help the review process. While these reports are written as official reviews, they were not requested for by the editor, and we leave it up to the editor and authors to use these reports to their advantage. While several students were asked to review the same paper, this was not done to provide the authors with much extra work. We hope that these reports will positively contribute to the scientific discussion and to the quality of papers published in HESS. This report was supervised by dr. Ryan Teuling.

In 1998, the Hurricane Mitch flooded the capital of Honduras with a return time of 500 years. This flood damaged 40% of the city's capital stock and one thousand casualties were reported (angel et al., 2004; JICA, 2002). Due to the power of floods, discharge and water level gauging equipment get destroyed. The discharge and water level of floods are needed for prevention and mitigation of floods. This high societal relevance initiated the authors to improve the knowledge of flood hydraulics. They state that post-event data have been used to simulate the flood hydraulics (Horrit et al., 2010; JICA, 2002), the GLUE framework has been used to account for uncertainty in models (Aronica et al., 1998; Brandimarte and Di Baldassarre, 2012; Pappenberger et al., 2005a, 2007) and that rainfall-runoff models have been coupled with hydraulic models (montanari et al., 2009). In this study the LISFLOOD-FP model was used to model the dynamics of the water level along the river channel and floodplain. This model scheme was specifically designed to predict flood inundation, and not flood routing (Bates, 2000). This model has extensively been validated in rural areas, but not in urban areas (Horrit, 2010). The use of a inundation-specific scheme can be justified, however this is not done in the paper, because the flood inundation was of greater importance in this report, having a compounded weight of 0.7 over the compounded weight of 0.3 for flood routing in the model evaluation. The use of this model over a full 2D dynamic model is justified by the fast computation time, which is favorable for the uncertainty analysis since the model will be run many times. For the uncertainty in the input hydrographs of the LISFLOOD model, 100 representative hydrographs were made. This was done with a GLUE analysis of the combination of TOPMODEL and Muskingum-Cunge-Todini. The choice of TOPMODEL is justified but the Muskingum-Cunge-Todini is not. However, due to the lack in cross section data in the subcatchments, it is logical not to use a more complex routing model (Todini, 2007). The simulations were concluded to be trustworthy because the simulated discharge, times of peaks and 90% of the high-water levels were within the uncertainty bounds of the evaluation data. In the introduction, the author states to combine a RRM, a hydraulic model and post-event data within an uncertainty analysis framework to prove that reasonable estimation of

an extreme flood is possible when hydrometric data are lacking. To my opinion, this aim is not ambitious enough. Previous work already have simulated this 1998 flood (ENEE, 1999; JICA, 2002) and the flood with a 50 year-return time design discharge in the same area (Mastin, M.C., 2002). This state-of-the-art is not included in the introduction, which I think should be the case. For the evaluation of the model, uncertainty bounds of the observed peak discharge, time of peak and water levels are introduced. Only the 50% of the observed peak discharge and not the 2.5 hours and 1.8 meter uncertainty bound of the time of peak and water levels are argued. Also it is not clear in the method how the extreme flood estimation is evaluated to be 'reasonable' or 'trustworthy'. To me it seems that after obtaining the results that 90% of all water level observations fall within their chosen uncertainty bound, they subjectively classified it as reasonable, trustworthy and realistic. Using the by the authors chosen uncertainty bounds, the aim of this paper was already reached by the MIKE11 simulations, the 1D unsteady flow hydraulic model from JICA, fourteen years before the start of this study (Figure 10 of the paper). Due to the unclear assumption of the uncertainty bound, the unambitious aim, and especially the fact that this aim was already reached by others, I advise to refuse the paper.

Major Arguments:

1. The aim to show that the 1998 flood could be reproduced is found in the abstract, the introduction and the conclusion. To me, the purpose of this study is not clear from this aim. In the discussion, the purpose is described: Bonnifait et al. (2009) reproduced an extreme flood event, but not within an uncertain analysis. In this study the uncertainties in model parameters, rainfall input and evaluation data have been accounted for. Not having this aim clear makes it hard to follow the storyline and what the actual message is that is meant to convey since there is no clear structure. In the AGU Fall Meeting Abstracts, the HEC-RAS model was compared with the Lisflood-fp model. This comparison resulted in better prediction with behavioural parameter sets and the obtained uncertain flood extension will be useful for decision makers (Fuentes Andino,

D. C., et al. 2012). Then in the EGU General Assembly Conference Abstracts in 2013, it is stated that the results of Lisflood will be evaluation with the GLUE analysis, but that the challenge is the how to evaluate the results when there are uncertainties in the model parameters and evaluation data. I think that both the aim of mapping the flood extension better and the aim of evaluating the Lisflood model were tried to be combined in this research, without this being possible. The results of the parameter sensitivity are analyzed (Figure 5-9), without this contributing to the aim and hypotheses. And the Lisflood model is compared with the JICA simulation (Fig 10), without really paying attention to the comparison. P14, l22 states in one line that the water levels of Lisflood encompasses the observations better than JICA, without ever stating that JICA produced water levels as well. This work could be a significant contribution to the society, if the uncertainty analysis produces improved flood-inundation maps compared to previous work, as stated in the abstract of Fuentes Andino in 2012. I suggest the aim to be: to prove that the flood water level and flood extension are more reliably modelled including than excluding an uncertainty analysis.

2. In the abstract of Fuentes Andino 2013 it was stated that the model evaluation is the challenge in this method. However, I do not see the argumentation of the choice of the fuzzy values included in this paper. In the model evaluation, the fuzzy set values were justified for peak discharge by selecting a value between the minimum and maximum uncertainty from literature of a mountainous area. However, the fuzzy values for the time of the peak and the water level are not justified. P14 L7 'The prediction of high–water marks was quite acceptable with average degrees of belief for the criteria ðÌŚŚ4−102 varying from 0.46 to 0.75 for behavioral simulations even when the criterion was relaxed.' Is 0.46 really quite acceptable? The a and b fuzzy set values are 0.5 and 1.8, meaning that with the degree of belief of 0.46, the model simulates 1,2 meter more or less than the observations. This sounds like a huge difference to me.

Minor Arguments:

1. P6,L21. 'Since discharge hydrographs were not measured. . . Chiquito River, Grande

River, Guacerique River, Salada Creek and Las Lomas Creek sub–catchments (points 1, 2, 5, 8 and 9 in Fig. 1 and 2 and Table1).' Why is this done? Because the hydrographs were not measured or to propagate the uncertainty of the input hydrographs in the GLUE analysis of the Lisflood model?

2. Header 3.2 states 'Rainfall-runoff modelling within an uncertainty analysis', while in fact already a combination of the TOPMODEL rainfall-runoff and the Muskingum-Cunge-Todini hydraulic model is implemented. I suggest: Calculating representative hydrographs for the subcatchments.

3. P12, L13: Here it is stated that the propagation of water level uncertainty is more evident at highly dense urban areas, referring to Figure 11. In Figure 11 I see a likelihood of inundation map. If the urban area is more likely to inundate, does this mean that there is more uncertainty here?

Literature Aronica, G., Hankin, B. and Beven, K.: Uncertainty and equifinality in calibrating distributed roughness coefficients in a flood propagation model with limited data, Adv. Water Resour., 22(4), 349–365, doi:10.1016/S0309-1708(98)00017-7, 1998. Brandimarte, L. and Di Baldassarre, G.: Uncertainty in design flood profiles derived by hydraulic modelling, Hydrol. Res., 43(6), 753–761, doi:10.2166/nh.2011.086, 2012. Bates, P.D. and De Roo, A.P.J., "A simple raster-based model for flood inundation simulation". Journal of Hydrology 236, p54–77, 2000. ENEE, "Modelacion Hidrologica y Hidraulica Cuenca Alta del Rio Choluteca". 1999. Fuentes Andino, Diana Carolina, et al. "Uncertainty estimation of simulated water levels for the Mitch flood event in Tegucigalpa." EGU General Assembly Conference Abstracts. Vol. 15. 2013. Fuentes Andino, D. C., et al. "Uncertainty estimation of water levels for the Mitch flood event in Tegucigalpa." AGU Fall Meeting Abstracts. Vol. 1. 2012. Mastin, M.C., 2002, Flood-hazard mapping in Honduras in response to Hurricane Mitch: U.S. Geological Survey Water-Resources Investigations Report 01-4277, 46. Montanari, M., Hostache, R., Matgen, P., Schumann, G., Pfister, L. and Hoffmann, L.: Calibration and sequential updating of a coupled hydrologic-hydraulic model using remote sensing-derived

water stages, Hydrol Earth Syst Sci, 13(3), 367–380, doi:10.5194/hess-13-367-2009, 2009. Pappenberger, F., Beven, K. J., Hunter, N. M., Bates, P. D., Gouweleeuw, B. T., 5 Thielen, J. and de Roo, A. P. J.: Cascading model uncertainty from medium range weather forecasts (10 days) through a rainfall-runoff model to flood inundation predictions within the European Flood Forecasting System (EFFS), Hydrol Earth Syst Sci, 9(4), 381–393, doi:10.5194/hess-9-381-2005, 2005a. Todini, E.: A mass conservative and water storage consistent variable parameter Muskingum-Cunge approach, Hydrol Earth Syst Sci, 11(5), 1645–1659, doi:10.5194/hess-11-1645-2007, 2007.

---

## Referee Comment (RC3) · Anonymous Referee #3 · 5 Dec 2016

"Reproducing an extreme flood with uncertain post-event information" by Diana Fuentes-Andino et al. deals with the possibility of modeling floods when there is a lack in data. The case considered is based on the flood event which occured in Teguciglapa (Honduras) due to hurricane Mitch. The purpose is to generate a probabilistic inundation map generated thanks to several modeling tools (TOPMODEL, LISFLOOD-FP) and considering uncertainty in parameters. The article is interesting and perfectly in the scope of the journal.

However, I think that this article might be fully improved.

The authors should highlight the novelties with their work and the difficulties. They should do a deeper analysis of the post-event data (quality and quantity), of the method used and of the results obtained.

[Figure]

This article is not self contained, it has to be read with other references. This is not very convenient, I recommend to give some information. For example, even if they are well-known some little details should be given about the different modeling tools which are used in this article (TOPMODEL, LISFLOOD-FP, GLUE, ...): equations, parameters, ...

The authors should mention some works which deal with Mitch event and its impacts in Honduras : - Westerberg, I., Walther, A., Guerrero, J.-L., Coello, Z., Halldin, S., Xu, C.-Y., Chen, D. & Lundin (2010). Precipitation data in a mountainous catchment in Honduras: quality assessment and spatiotemporal characteristics. Theor Appl Climatol, 101. 381-396. - Mastin, M.C. and Olsen, T.D. (2002). Fifty-Year Flood-Inundation Maps for Tegucigalpa, Honduras. U.S. Geological Survey Open-File Report 02-261. - Haile, A.T. (2005). Integrating Hydrodynamic Models and High Resolution DEM (LI-DAR) For Flood Modelling. International Institute for geo-information science and earth observation Enschede, the Netherlands.

All the hydraulic parameters do not have the same impact and the same influence on the results of the modeling. There are some good references on this topic in litterature, they have to be mentioned and some words have to be said. Because all parameters cannot be considered of equivalent value. This remark is also valid for the data. A more in depth discussion should be done on this topic which one of the main points of the article. For example discharge and roughness coefficient are strongly connected, so uncertainties on discharges should impact strongly the roughness coefficient ...

In both the introduction and the section 3. "Method", for the uncertainty analysis aspects, the focus is given exclusively on the GLUE method which is used here applied in the field for a couple of years already. However it would be very welcome to contextualize the interest of using the GLUE method within the framework of more recent methods applied in the field of uncertainty analysis in recent years : e.g. in hydraulic modeling in 1D see (Bozzi et al., 2015) and in 2D see (Willis, 2014) which applies a screening method in 2D and lastly (Abily et al., 2015 & 2016) for global sensitivity analysis applications in 2D and spatialisation of uncertainty aspects. I recommend the

authors to provide a short subsection/paragraph which makes the synthesis of this type of approaches explaining what is the place of GLUE. Obviously above mentioned approaches are computationnally costly, but a stte of the art is clearly lacking in this article to enhance added value and limits of what is done by the authors on the uncertainty aspect compare to what is existing in literature and to put it in perspective for the readers especially for those who are not familiar with uncertainty analysis. I recommand in this topic : - Abily, M., Bertrand, N., Delestre, O., Gourbesville, P., & Duluc, C.-M. (2016). Spatial Global Sensitivity Analysis of High Resolution classified topographic data use in 2D urban flood modelling. Environmental Modelling & Software, 77. 183-195. - Abily, M., Delestre, O., Amossé, L., Bertrand, N., Richet, Y., Duluc, C.-M., Gourbesville, P. & Navaro, P. (2015). Uncertainty related to high resolution topographic data use for flood event modelling over urban areas: toward sensitivity analysis approach. ESAIM: Proceedings and Surveys, 48, 385-399. - Bozzi, S., Passoni, G., Bernadara, P., Goutal, N. & Arnaud, A. (2015). Roghness and Discharge Uncertainty in 1D Water Level Calculations. Environmental Modeling & Assesment, 1-11. - Willis, T.D. (2014). Systematic analysis of uncertainty in flood inundation modelling. Doctoral dissertation, University of Leeds. - Iooss, B. & Lemaître, P. (2015). A review on global sensitivity analysis methods. Uncertainty management in Simulation-Optimization of Complex Systems: Algorithms and Applications, Ed. C. Meloni and G. Dellino, Springer.

Here are some comments/questions on details : - p.7 l.19 "combining the rainfall-runoff TOPMODEL", as mentioned before, some details should be given on the modeling tools especially TOPMODEL, in order to show that it is also a model. Because some modeling tools such as HEC-RAS, MIKE11, ... are based on different physically based models/equations and the word model cannot be used for these tools. So it should be clarified if TOPMODELis based on one model and might considered as a model. If not, the sentence should be changed into "combining the rainfall-runoff results generated by TOPMODEL". - p.8 l.15 "with channel roughness coefficient (nCU) assumed uniform along all the reaches." it should be told if it is a reasonnable asumption. - p.8 from line 17 to 25, I found this paragraph well written. - p.9 line 5-6 Are a and b percentages

or hours? It has to be clarified here p.10 l.26-27. - p.9 l.28 "a downstream boundary condition" it should be told which kind of boundary condition is used. I guess it is water level/height. - p.10 from line 5 to 12, are the choices given reasonable, it should be discussed a little to justify these choices. - p.10 l.21 I think that "Two degree of belief values" should be changed into "Two degrees of belief values". - p.10 l.23 the same remark for "Ninety-nine degree of belief values". - p.10 l.28 "metres" should be changed into "meters". - p.10 l.31 "... degrees of belief." should be changed into "...degrees of belief:". - p.10 l.32 after formula (1) a comma should be added. - p.11 l.1 "Where ..." should be changed into "where ...". - p.11 l.4 "to all the observed maximum water level" into "to all the observed maximum water levels". - p.11 l.23 "... 47 894 out of 130 000" something is missing: "130 000 behavioural simulations"? - p.12 line 3 to 10 are the behaviors due to the change of parameters described in this paragraph expected? It should be justified a little. - p.12 l.16 "4619mˆ3 sˆ-1" a space should be added between the value and the unit. - p.13 l.4 "an RRM" should be changed into "a RRM". - p.13 l.5 "was proven" should be changed into "was proved". - p.13 l.8 "... be used for forecasting ...", it should be told what is forecasted. - p.14 l.21 "The LISFLOOD-FP model ..." same remark as previously, it should be told why LISFLOOD might be considered as a model, if not it should be changed into "The LISFLOOD-FP modeling tool" or "The LISFLOOD-FP software". - p.14 l.30 "DEM" the meaning of DEM has not be given. - p.15-16 conclusion, as told before the challenges/difficulties and the novelties have to be highlighted.

Figures/graphics should be improved.

As a conclusion, I would say that this article is interesting and is in the scope of the journal. It needs major revision. Once all points would be fixed, it should be a very interesting article for the community.

———————————————

---

## Author Comment (AC1) · 12 Dec 2016

We thanks A. Tegos for carefully reading and for his valuable comments to our work. We will include in the introduction section insights of previous works on probabilistic inundation maps including those recommended by the reviewer. We are especially grateful for introducing the work of Dimitriadis et al. (2016) that we were not aware of. We have studied the work of Dimitriadis et al. (2016) and found it important to refer in our work.

References

Dimitriadis, P., Tegos, A., Oikonomou, A., Pagana, V., Koukouvinos, A., Mamassis, N., Koutsoyiannis, D. and Efstratiadis, A.: Comparative evaluation of 1D and quasi-2D hydraulic models based on benchmark and real-world applications for uncertainty assessment in flood mapping, J. Hydrol., 534, 478–492, doi:10.1016/j.jhydrol.2016.01.020, 2016.

---

## Author Comment (AC2) · 12 Dec 2016

We thank Referee # 1 for her/his interesting observations that will help us to improve our manuscript.

In general we agree with the reviewer and we will developed more in the discussion sections on the implication of the results from the data, hydrological and hydraulic perspectives.

More specifically, we addressed each of the reviewer points as follows:

1) We agree that uncertainties in discharge estimates will directly affect estimates of roughness coefficients and vice versa (Aronica et al., 1998; Warmink and Booij, 2015; Wohl, 1998), therefore a localized setting for the roughness coefficient can improve

[Figure]

model fit (e.g. Romanowicz and Beven, 2003). With the data availability in our study, we could estimate the localized roughness coefficient for the most downstream reach, using points 3, 6 and 7 in Fig. 1 and 2, however this will prevent us from using point 7 for validation. In addition, except for the most downstream reach, for the other four reaches we only have observed high water marks to calibrate against. Thus, a localized roughness-coefficient will lead to an increased number of parameters in the hydraulic model when we do not have enough information at each reach to constrain the local roughness. We will include in the manuscript a discussion on the interrelationship of roughness coefficient with discharge and about the improvements that can be done in this work if more data was available. Thus, we will also suggest strategies for improvements on the post-event data collection campaigns.

2) We will add and explain better this part of the procedure in the manuscript based on the following comment. Regarding time of the peak, all observations have the same source i.e. witnesses account. Regarding the peak discharge, points 1, 2, 3 and 5 at fig. 1 and table 1 were used for the calibration. The uncertainty at 1 and 2 were chosen considering, and assumed larger, than the values suggested at Benson and Dalrymple (1967) and Cook (1987). The discharge at point 5, although it was estimated by running a RRM by JICA (2002), was used for calibration here since its magnitude was similar to the maximum peak outflow measured at Los Laureles dam which is located in the same river and with nearly same contributing upstream areas than point 5 in Fig. 1. The uncertainty given to points 1 and 2 was considered large enough and was also chosen as the uncertainty at point 5 because there was no better source of information to constrain it. Thus, the peak discharge at Los Laureles reservoir was used to support our decision to use the discharge produced by JICA (2002) at point 5 to constrain maximum peak discharge at that point. The hydrograph at the Concepción reservoir was routed using the TOPMODEL and the Muskingum–Cunge–Todini routing approach.

3) We checked for consistency of the data prior to the calibration procedure. We will

incorporate that into the manuscript, including a discussion on the implications of the quality and the quantity of the data.

From the analysis we did prior to the calibration procedure we found that maximum peak discharge and time of the peak information was consistent. Consistency in the high water marks was also checked, and we did notice that the profiles for the observed high water marks were not smooth (some sudden jumps without any obvious physical explanation) but this is expected given the origin of those observations (witness accounts). Thus we did not eliminate any of the observations but instead assigned a reasonable uncertainty range to each of them for the calibration procedure. Because of the quantity of high water mark observations (available on average every 200 meters, location in Fig. 2) it was not convenient to assign numbers for each of the observations.

4) We agree with the reviewer comment, the global calibration procedure is flexible p. 14 Lines 10-12: "The weights could be changed according to the purpose of the study which might also result in different ensembles being behavioural for different purposes (Pappenberger et al., 2007)". Thus, results from a sensitivity analysis will be bound to the weighting scheme chosen. We did not look into differences arising from different schemes as we wanted to have a restricted focus within this work.

5) To parameterize the TOPMODEL and the Muskingum–Cunge–Todini routing in our work, we included the often considered parameters within these modelling concepts and the parameters that we thought would be important in driving the hydrograph in our catchments. We avoided to constrain our model from the start by simplifying it. Instead, to consider a possible over-parametrization, we check for stability in the produced cumulative distribution of the output predicted variables (i.e. peak discharge and time of the peak) and made as many simulations until the distributions stabilized (the distributions did not changed after adding more realisations (p 8 Lines 19 to 25). Thus even if the RRM model was over-parametrized, the stability of the obtained cumulative distribution indicated that the parameter space was explored sufficiently. Within the LISFLOOD hydraulic model, besides the roughness coefficients, we added a slope for

the downstream boundary condition as it is known to affect the predicted water level at the downstream end (Pappenberger et al., 2006) especially because we made a simple normal flow assumption at that boundary. Prior to the setting of the hydraulic model, we explored its sensitivity to the channel depth and channel width values (through a multiplying factor), which lead to treating the channel width as an uncertain parameter in the analysis. Thus we finally considered a total of four parameter for the hydraulic modelling which we think is a reasonable number. The ranges chosen to sample the model parameters were set wide enough to consider uncertainty in parameter values especially when they represent spatially-aggregated effective values. Using a range to represent the uncertainties each parameter is associated to, can lead to a large variation in the resulting range of figures 5-7. We tested if the available data can help to constrain and decrease those ranges. We will add value to the manuscript by incorporating a framework for the GLUE within as an uncertainty analysis technique and explain the reason why we chose GLUE methodology.

6) We considered a rainfall multiplier to represent uncertainty in the spatial average estimation of rainfall as it has shown to improve model prediction (Fuentes Andino et al., 2016), this uncertainty is important to consider when few rain gauges exist as in this study. As the multiplier is considered as an extra parameter, it interacts with all the model parameters, as the reviewer suggests. Note that in the face of such epistemic uncertainties, the GLUE formulation used allows for complex interactions between parameters (not just variance/covariance forms) in that it is the parameter SET that is evaluated as behavioural or not. Thus the complex interactions are contained implicitly in the resulting ensemble of behavioural simulations. As the reviewer pointed out, the limitations due to the use of few rainfall gauges, was evident specially for predicting the time of the peak. We will further highlight this limitations and discuss the importance of a denser rain-gauge network.

7) a) We think to improve the work by extending on the discussion section about the inconsistency between model and observations at some specific locations. b) It was

not possible to identify inconsistency in the post-event estimated peak discharge, but we noticed after the simulations: "as in comparison to the Grande and Guacerique sub–catchments, most of the hydrograph simulations for Chiquito River sub–catchment were rejected because the simulated peaks were larger than the observations (even considering the uncertainty) (Fig. 5)" (P 14 L33 to P15 L 1-3) this was only possible to see after the simulations. All major revision we consider to make in the manuscript are summarized in the Major revisions document uploaded as supplement.

References

Aronica, G., Hankin, B. and Beven, K.: Uncertainty and equifinality in calibrating distributed roughness coefficients in a flood propagation model with limited data, Adv. Water Resour., 22(4), 349–365, doi:10.1016/S0309-1708(98)00017-7, 1998.  Benson, M. A. and Dalrymple, T.: General field and office procedures for indirect discharge measurements, in Techniques of Water- Resources Investigations of the United States Geological Survey.  [online] Available from: http://pubs.usgs.gov/twri/twri3-a1/html/pdf.html, 1967. Cook, J.: Quantifying peak discharges for historical floods, J. Hydrol., 96(1–4), 29–40, doi:10.1016/0022-1694(87)90141-7, 1987. Fuentes Andino, D., Beven, K., Kauffeldt, A., Xu, C.-Y., Halldin, S. and Baldassarre, G. D.: Event and model dependent rainfall adjustments to improve discharge predictions, Hydrol. Sci.  J., 0(ja), null, doi:10.1080/02626667.2016.1183775, 2016.  JICA: On flood control and landslide prevention in Tegucigalpa metropolitan area of the republic of Honduras, Pacific consultants International and Nikken consultants, Tegucigalpa, Honduras., 2002.  Pappenberger, F., Matgen, P., Beven, K. J., Henry, J.-B., Pfister, L. and Fraipont de, P.: Influence of uncertain boundary conditions and model structure on flood inundation predictions, Adv.  Water Resour., 29(10), 1430–1449, doi:10.1016/j.advwatres.2005.11.012, 2006. Pappenberger, F., Beven, K., Frodsham, K., Romanowicz, R. and Matgen, P.: Grasping the unavoidable subjectivity in calibration of flood inundation models: A vulnerability weighted approach, J. Hydrol., 333(2–4), 275–287, doi:10.1016/j.jhydrol.2006.08.017, 2007. Romanowicz, R. and Beven, K.:

[Figure]

Estimation of flood inundation probabilities as conditioned on event inundation maps, Water Resour. Res., 39, 12 PP., doi:200310.1029/2001WR001056, 2003. Warmink, J. J. and Booij, M. J.: Uncertainty Analysis in River Modelling, in Rivers – Physical, Fluvial and Environmental Processes, edited by P. Rowiński and A. Radecki-Pawlik, pp. 255–277, Springer International Publishing., 2015. Wohl, E.: Uncertainty in Flood Estimates Associated with Roughness Coefficient, J. Hydraul. Eng., 124(2), 219–223, doi:10.1061/(ASCE)0733-9429(1998)124:2(219), 1998.

---

## Author Comment (AC4) · 12 Dec 2016

We are thankful to all referees and others for their valuable comments to our work. All the comments and suggestions done will greatly help us improve our manuscript. After going through all of them, the major improvements in this work will be:

1. The uncertainties in the data, their limitation and implication in the results will be further discussed in the revised version. For example, we bring up the sources of uncertainties introduced by the roughness coefficient and its interaction with discharge.

2. We will developed more in the discussion sections on the implication of the results from the data, hydrological and hydraulic perspectives. Also we will develop more on the possible reasons of disagreement for some of the maximum water level observations.

[Figure]

3. A literature review about other available methods for uncertainty analysis and more details on the choice of the Generalized Likelihood Uncertainty Estimation (GLUE) method in our work will be included in a revised version.

4. We will make the manuscript more self-contained by adding more description and incorporating appendices of the models and tools used, we will also further explain reasons for the assumptions and decisions done throughout the work.

5. The value of this work will be increased after adding suggested literature related with the impact of the hurricane Mitch and of the quality of the data in the region.

6. We will discuss and suggest strategies for improvements on the post-event data collection campaigns.

7. For clarity, the presentation of some figures, especially Figures 10 and 11, will be improved.

8. The references will be updated including a link to the work from JICA (2002): http://libopac.jica.go.jp/search/detail.do?rowIndex=7&method=detail&bibId=0000054206

9. The clarity and structure in the manuscript will be improved, we will also highlighting the novelty of this approach.

---

## Author Comment (AC5) · 12 Dec 2016

We thank reviewer 2 for his/her constructive remarks, which will help us improve the final manuscript. We especially note the request for an improved presentation of the material. We refer the reviewer to the summary in the "Major_revision", added as one of the comments, where we summarized all major revisions that will improve this work. More specifically, we will make use of all suggestions of reviewer 2 when revising this manuscript in what follows:

1. We will also improve the descriptions of each models and tool used and their underlying assumptions. For example we will provide more description for the LISFLOOD-FP, TOPMODEL and the Muskingum-Cunge-Todini routing. Explain further on the K-means algorithm and reasons we used the Kuiper test.

[Figure]

2. The content and structure of the manuscript will be improved so to make it clearer.

3. The content of the manuscript will be improved by discussing more on the different sources of uncertainties and their implication in the results. For example, we bring up the sources of uncertainties introduced by the roughness coefficient and its interaction with discharge.

4. We will add a literature review about other available methods for uncertainty analysis and explain the reason to choose the Generalized Likelihood Uncertainty Estimation (GLUE) method in our work.

5. We will expand more and clarify the discussion section, especially when it comes to the possible reasons of disagreement for some of the observations.
* * *

---

## Author Comment (AC6) · 12 Dec 2016

We thank Anouk Sprong for suggested literature and for providing constructive comments, which will help us improve the description of our scientific work. All remarks will be carefully considered and, as suggested, for example, we will revise figures and clarify our procedures in the revised paper.

First, we would like to specifically address two key questions that were pointed out by Anouk Sprong.

Comment: In my opinion the results of Bonnifait et al. (2009) and Fuentes-Andino are more or less the same. So the question arises what is the added value of the GLUE framework to the modelling approach of combining a RRM with a hydraulic model.

Response: Within the uncertainty analysis framework presented in our manuscript, to account for the parameter uncertainty many possible parameter combinations were generated. Bonnifait et al. (2009) considered fewer parameter sets. Both approaches are valid to reflect the uncertain nature of the system, but the meaning of the resulting predictive uncertainty is different for the reason reported above.

Comment: Smith et al. (2002) did also research about the peak discharges caused by Mitch. Using standard USCG techniques, he found discharge value within the range proposed by Fuentes Andino for the same reaches. So why using model techniques while standard USGS techniques give more or less the same result concerning the discharge.

Response: Peak discharge obtained in Smith et al. (2002) were relevant to this study to calibrate the hydrological model so to obtain a hydrograph instead of a peak discharge as input for the hydraulic model. Through that it was possible of, instead of a peak, to propagate the hydrograph thus acknowledging the unsteady nature of the flow, which is crucial not only for hydraulic simulation, but also for the design of flood protection structures (e.g. flood retention reservoirs).

For the other comments, we agree with Anouk Sprong that validation is an important part of the modelling process. Within the limitation of data availability in this work, we compared our results with those produced by JICA (2002) which used a different modelling technique. There was no information about flood extent available to be used for this purpose. The proposed methodology can make use of the Bayesian concept, new data, including post–event, can be used to update the identified parameters.

Also, areas contribution to points 8 and 9 are large enough so the inflow has to be considered for the upstream boundary conditions. As there are no observations for those points, the hydrograph was inferred using behavioural parameters (P8 L28-30). Since those areas were smaller comparing to the rest of contributing area (at points 1, 2 and 5), it is expected a limited impact on the results (P13 L30-33 and P14 L1-3). We

acknowledge that this was not clear in the original manuscript. Thus, we will clarify this point.

Moreover, we will better explain what we mean by "propagation of the water level uncertainty in the flood extent was more evident at highly dense urban areas".

We will explain that the time of the peak was not surveyed/available for all the points in Table 1.

We think that details of what we want to show might decrease if we expand to show more of the area in Fig. 1.

Anouk Sprong can also refer to the supplementary document "Major_ revision" including a link where the work of JICA (2002) is available for download. The same document also describes improvement of figures as well as description of models and tools. Please, refers to the points 1, 4, 7, 8 and 9 of that document related to some of the comments in this review.

References

Bonnifait, L., Delrieu, G., Lay, M. L., Boudevillain, B., Masson, A., Belleudy, P., Gaume, E. and Saulnier, G.-M.: Distributed hydrologic and hydraulic modelling with radar rainfall input: Reconstruction of the 8–9 September 2002 catastrophic flood event in the Gard region, France, Adv. Water Resour., 32(7), 1077–1089, doi:10.1016/j.advwatres.2009.03.007, 2009. JICA: On flood control and landslide prevention in Tegucigalpa metropolitan area of the republic of Honduras, Pacific consultants International and Nikken consultants, Tegucigalpa, Honduras., 2002. Smith, M., Phillips, J. and Spahr, N.: Hurricane Mitch: peak discharge for selected river reaches in Honduras, U.S. Geological Survey. [online] Available from: http://pdf.usaid.gov/pdf_docs/Pnacp984.pdf, 2002.

---

## Author Comment (AC7) · 12 Dec 2016

We are grateful to Ilja America for reviewing our manuscript, for summarising our work and for her valuable comments which will greatly help us improve our manuscript.

Response to Major arguments

Hypothesis:

-We agree with reviewer that the hypothesis can be better formulated to make it clearer. We will consider the reviewer's suggestion. Thanks.

Model evaluationïijŽ

-Thanks for the reviewer suggestions on Liu et al. (2016) and other flood hazard assessment methods that can be interesting to try. In our work we wanted to explore the

possibilities to use post-event data, a combination of models and uncertainty analysis to assess flood modelling in data scarce areas, we are sure that there are many other approaches that are worth testing. We will discuss this issue in the revision.

We are aware of the shortcoming regarding validation, the proposed methodology can be seen as a learning strategy, part of the Bayesian concept, new data, including post–event, can be used to update the identified parameters. We will discuss this issue in the revision.

Uncertainty:

-We will improve the text in our manuscript by adding and clarifying better some of the points raised in the comments: -Extend on the discussion of the uncertainties associated with the data (see for more detail in the Major_revisions in the supplementary documents).

-Regarding point 1 on comparability between the different post-event data, the reviewer can refer to point 2 and 3 in answer provided in the answer to reviewer 1.

-Regarding point 2, on the number of parameters used, refer to point 5 provided in the answer to reviewer 1 and point 1 in the Major_revisions.docx.

Minor arguments:

-All the minor arguments will be considered by clarifying and or correcting into the new version of the manuscript.

-In response to the comment: "The TOPMODEL parameter sets from Grande and Chiquito River sub-catchments were used to simulate the hydrographs at Salada creek and Las Lomas creek subcatchments (P11, L10-13). The reason for this is missing". Drainage areas of points 8 and 9 are large enough so the inflow have to be considered for the upstream boundary conditions. However, there were no stage records available for those points and the hydrograph were estimated using behavioural parameter from neighbouring catchments P8 L28-30, those areas were smaller compared to the rest

of contributing area (at points 1, 2 and 5) (at points 1, 2 and 5) that it was expected not to greatly impact the results, which was the case (p13 L30-33 and P14 L1-3). We will make this clearer in the revised version.

Minor issues:

-Minor issues, regarding the comment on the abstract, the role of the rainfall, JICA Report, improvements in Figure 11 will be incorporated into the new version of the manuscript. More specific points:

-In response to the comment: "Fig 3 : The figure is not very valuable for understanding the story. Therefore it is not needed": Figure 3 shows the peak in the rainfall so as to compare it with the resulting peak in the propagated hydrograph, showing thus the importance of rainfall in driving the models p14 L 15-20.

-In response to the comment: "P35, Fig 11 : The figure is very unambiguously. If there is a likelihood of inundations than this likelihood is mainly 0.8-1. This is very peculiar": Likelihoods at Fig. 11, varied from 0 to 1 because the global score for all behavioural sets were scaled by a constant C, to sum one (Beven, 2009) (P7 L15-17).

The reviewer can also refer to the supplementary document "Major_ revision", specifically points 1, 3, 4, 8 and 9 address some of the points raised by the reviewer.

References

Beven, K. J.: Environmental Modelling: An Uncertain Future? An introduction to techniques for uncertainty estimation in environmental prediction, Routledge: London., 2009. Liu, R., Chen, Y., Wu, J., Gao, L., Barrett, D., Xu, T., Li, L., Huang, C. and Yu, J.: Assessing spatial likelihood of flooding hazard using naïve Bayes and GIS: a case study in Bowen Basin, Australia, Stoch. Environ. Res. Risk Assess., 30(6), 1575–1590, doi:10.1007/s00477-015-1198-y, 2016.

---

## Author Comment (AC8) · 12 Dec 2016

Thanks to Kenechukwu Okoli, for his summary and insight about the use of the fuzzy set theory. We will improve explanation in our text regarding the choice of the membership functions.

Our comment in our choice of a trapezoidal membership:

We used a trapezoidal membership function to represent uncertainty in the post-event surveyed high water marks, because the observed values, is derived from peoples' memory of the event, thus we considered those observations not to be crisp estimations. Thus we used a range to represent a most likely error in those values so to allow a high likelihood within that error interval.

---

## Author Comment (AC9) · 12 Dec 2016

We thanks Timo Kelder for providing constructive comments that will help us improve our manuscript. All remarks will be carefully considered. We also refer Timo Kelder to the supplementary document "Major_revision", where a summary of all major revisions after all reviews comments are presented. Points 1, 2, 4, 5 and 9 of the document are related to the some of the comments raised by the reviewer.

More specifically, we will like to address some of the comments as follows:

We agree on the benefits of using the LISFLOOD model for flood-extent predictions (Bates et al., 2010; Horritt and Bates, 2002). This does not exclude the possibility to use it for flood routing.

[Figure]

We appreciate the suggestion to justify the choice of modelling tools used here, thanks for pointing out the benefit of the Muskingum-Cunge-Todini approach; we will improve this in our work. Please refer to point 4 in Major_revision .doc.

The work will benefit by including the suggested references: Mastin and Olsen (2002).

Major arguments:

1. We will include the reviewer suggestion to make clearer the manuscript (also refer to point 9 in Major_revision.doc).

2. We will improve our argument concerning definition of acceptability of the modelling results (also refer to point 1 in Major_revision.doc).

We will address each of the points in Timo Kelder's Minor Arguments, specifically:

Point 1: we will improve the text regarding the reason why hydrograph were estimated.

Point 2: We thank Timo Kelder for his suitable suggestion to improve the header 3.2.

Point 3: We will clarify the text, explaining how the spatial variability of the topography in the urban areas makes model results more sensitive in these areas.

References

Bates, P. D., Horritt, M. S. and Fewtrell, T. J.: A simple inertial formulation of the shallow water equations for efficient two-dimensional flood inundation modelling, J. Hydrol., 387(1–2), 33–45, doi:10.1016/j.jhydrol.2010.03.027, 2010. Horritt, M. S. and Bates, P. D.: Evaluation of 1D and 2D numerical models for predicting river flood inundation, J. Hydrol., 268(1–4), 87–99, doi:10.1016/S0022-1694(02)00121-X, 2002. Mastin, M. and Olsen, T.: Fifty-year flood-inundation maps for Tegucigalpa, Honduras, U.S. Geological Survey., 2002.

---

## Author Comment (AC10) · 12 Dec 2016

We thank Referee # 3 for her/his feedback and constructive comments, which will help us improve the description of our scientific work. More specifically, we agree with the reviewer's criticism, as some points were not sufficiently well described. Thus, we will address all comments by revising our paper, as follows:

- Highlight the novelty of this scientific work in view of the state of the art.

- Incorporate in the manuscript a critical discussion of the quality and the quantity of the data and their implications.

- Make the manuscript more self-contained by adding more information on the models and tools used. For example, a better description of the LISFLOOD and appendices to

TOPMODEL and Muskingum-Cunge-Todini routing will be incorporated. We will also explaining the assumptions and decisions done throughout the work.

- Add more literature about Mitch impact as well as the quality of the data in the region, thanks for the recommended literature.

- The content of the manuscript will be improved by discussing more on the different sources of uncertainties and their implication in the results. For example, we bring up the sources of uncertainties introduced by the roughness coefficient and its interaction with discharge.

- Include reference to other methods for uncertainty analysis, and explain the reason why we use the Generalized Likelihood Uncertainty Estimation (GLUE) method in our work. We will consider the reviewer suggested literature in this field, thanks.

- Improve the figures, as suggested.

- Amend typos and other address all minor, editorial comments.

Please note that all major changes we consider to make in the manuscript are summarized in the Major_revisions (uploaded as a comment in the discussions).

―――――――――――――――――

---

## Author Response (AR2)

**Response by Fuentes-Andino et al. (HESS)**

**Comment from Editor**
*The revised version of the paper was sent for a round of review. The Reviewers #1 and #3 are totally satisfied with this version and asked for minor revisions, while Reviewer #2 asks more in depth analysis to enrich the proposed paper. The paper will be reconsidered after "revisions". In the revised version, authors are invited to respond point by point to each comment of each reviewer, mainly Reviewer #2.*

**Reply to Editor**
We thank again the Editor, Dr. Roger Moussa, and the three Anonymous Referees for this second round of review. This point-by-point document describes the way in which all remaining comments were carefully addressed in the second revision of our document.

We believe that this process has allowed us to improve the description of our work.

We updated the manuscript according to all comments (specified below in the answers to each of them). In addition to the Reviewers' suggestions we did the following updates:

- The catchment average slopes were changed (P.5 l.22) in section 2.1 on the description of the area. Previously we used the method described in Di Lazzaro (2009) to estimate the average slope in the catchment i.e. the average of the ratio between the drop in height in each cell in the basin to the outlet and the distance between them following the direction of the flow. Now the more commonly method of average of maximum rate of change from each cell to its neighbours was used.
- Add more details, to clarify better Eq. A5 (P.23 l.25).
- In general we made small improvements in the writing of the article.

**Comments from Referee #1**
*Several modifications and additions are included in the second version of the manuscript, and described in an accompanying note. But, these modifications are mostly minor, many comments of the reviewers where acknowledged and even included in the manuscript, but not really answered. Basically the results and their analysis remain unchanged, where my feeling is that a more in depth analysis was necessary to enrich the proposed paper.*

*I am finally hardly overall convinced by the content of the manuscript. It was the sense of my numerous question and comments in my initial review. The authors focus on the technique and its implementation and seem to forget, doing this, what is the real objective of such a post event analysis: learn from the observations about the underlying processes. If we consider that the calibrated model parameter values summarize what has been learnt from the confrontation between observations and models, the conclusion of the study should be that very little has been learnt. The initial uncertainties about the parameter values could hardly be reduced (fig. 6 and 7). No surprise considering the scarcity of the available set of observation (even the rainfall amounts are relatively unknown). Presented in this way, this result is really disappointing. It is of course due to the scarcity of the available data set but also to the method used. Calibrating all the models and all their parameters using all available data, everything put in a single "pot" without further strategy, without separating and sorting out the various questions (runoff production, flood wave propagation, stage-discharge relations), without a real critical analysis of the available data, could hardly lead to a different result.*

*In do not share the statement of the authors that the method proposed could be taken as a reference example.*

*The paper is well written, but I can finally hardly imagine what its real contribution to the hydrological or hydraulic science is. Roughly said, it shows that a large variety of combinations of parameters may produce "behavioural" results in a largely over-parameterized case study. No surprise at all. I do not really see what other sort of conclusion can be drawn from the presented approach. Since the models could not really be calibrated, I do not share the final the conclusion that the presented work could be of any help for useful planning and flood forecasting. The application of the Glue method does not hide that the manuscript is relying on much too scarce, inaccurate data set, furthermore analysed using a much too coarse approach.*

**Reply to Referee #1**
We thank Referee # 1 for her/his feedback and comments.

We considered all the comments received in the first revision, the manuscript was updated considering many of the suggestions. If a suggestion was no considered, the reason was argued in the answer given to each of the reviewers (found in the discussions for this manuscript).

We want to point out that we are presenting a methodology not just an application. This methodology deals with the problem when both flow–runoff and flow–routing need prediction under data quantity–and–quality problems. Thus, more than learning from models, we aim to show what is possible to do considering those data limitations (which needs to be addressed in the literature).

It is not expected that all parameters in a model are equally sensitive as shown in Figures 7 and 8 (for example) and in many other previous works that involve uncertainty analysis. Different sensitivity in different model parameters occurs, even when data availability is not a problem, because of the various sources of uncertainty (i.e. equifinality thesis (Beven, 2006)). The reduction of the uncertainty in the marginal distribution for some of the model parameters led to a reduction in the predictive uncertainty (Fig. 6). Thus, if anyhow we need to predict, it is preferred to use the black hydrograph from Fig. 6 (constrained by using post–event data) than the wide range (grey ones) obtained when not information of the event is available. Similarly, the use of all produced hydrograph as input to the hydraulic model would have led to a much larger predictive uncertainty of the flood extend, likely useless for planning or parameter calibration. Thus, constraining of the model parameter within this methodology helped to reduce the predictive uncertainty.

 We acknowledge the usefulness of a sensitivity analysis for identifying parameters that are more important in driving the outputs from a model (P.4.15-20), however this can only be done (and has been done) when more quantity and a better quality of data is available.

After the reviewer's comments, more effort was put to highlight what can be learnt and gain from confronting post–event–estimated observations and models within the methodology proposed here:

- Reduction in the predictive uncertainty (as pointed out above). In addition we discussed that uncertainty can be further reduce and that the reliability of identified parameter sets (for forecasting) can be improved by using the present methodology within a Bayesian framework (P.2 l.5-6, P.20 l.24-25 and P.22 l.7-8).

- To identify locations, within a specific study area, where additional data would be required to understand the reasons of disagreement (having already considering the uncertainties) between model and observations. Such locations are useful to improve model structure or post–event–data estimation methods (see e.g. P.19 l.6-8, P.19 l.22-26 and P.19 l.27-30 to P.20 l.1) and in our discussion and conclusions in P.20 l.4-5, P.20 l.25-27 and P. 22 l.2-5.

- It was possible to assess the lack of spatial representation of the rainfall (as pointed out by the reviewer) that prevented a fair prediction of the time of the peak. Thus, from this work it was possible to conclude on the importance of rainfall data for reproducing an unmeasured flood event. Thus, more effort to collect rainfall data should be placed when discharge measurement are more difficult to obtain during extreme events (P.18 l.31-32 to P.19 l.1-5).

- In general, by making sense of all available data (even if it was limited) and the possible processes, we can learn from the modelling system. Such learning is expected to be expanded when more data, through more events, become available.

Further points to learn from the present methodology:

- This methodology deals when extreme large events (like the case of the hurricane Mitch) impact a vulnerable location when little or no data is available. Thus our methodology was oriented, and can only be applied, to the calibration of extreme events under data limitations. There are no works in literature approaching this problem through the use of rainfall–runoff, hydraulic-routing and uncertainty analysis as done here. We are proposing just one methodology; however the scientific community would benefit from the development of more.

- We proposed a methodology to consider the sudden release of water (e.g. from a dam) in a rainfall-runoff model (RRM ) by combining the TOPMODEL with the Muskingum–Cunge–Todini routing (MCT). This method is of interest because the MCT is adapted to be use when no topography of the cross section is available. In addition, the TOPMODEL and MCT combination might be an option to have a more localised model parameterisation (for those cases of richness in data) at the main channel while keeping a more compounded representation of the parameters from the RRM in the upstream area. Still the method needs further testing (P. 21 l.25-28).

In the present methodology, we strategically separated the rainfall–runoff from the flow–routing calibration (sections 3.4 and 3.5), we did not have information of the stage–discharge relationship, thus we did not considered this in the analysis but we discussed it, as already pointed out in our answer to the first revision.

We do not understand the reviewer's comment: "Since the models could not really be calibrated". We carried out a calibration of a model within an uncertainty analysis framework.

As stated by the reviewer, data scarcity and in addition epistemic uncertainty (see. Beven, 2016) is a reason to not use a formal statistical approach, such as the Bayesian, for the uncertainty–analysis (this was explained in our answer from the first revision).

**Comments from Referee #2**

*Authors have done a great job and I appreciate their efforts. Authors received numerous remarks of various kinds and it was kind of an art to find a balance among them. In principle most of the remarks pertained to the clarity of the presentation of methods and results and in my opinion the quality of the description was significantly improved. Authors complied with almost all suggestions of the referees and other commentators. It made the paper a bit lengthy but it was definitely caused by the remarks and expectations of the referees. I am satisfied with the proposed changes and I recommend the paper for publication in the present form.*

**Reply to Referee #2**

We thank Referee #2 for her/his going through our work and providing valuable comments.

**Comments from Referee #3**

*The authors describe a way to reproduce an extreme flood event due to hurricane Mitch in Tegucigalpa in Honduras thanks to TOPMODEL and LISFLOOD-FP modelling tools. The problematic and the methodology are well explained. The points of the conclusion are clear. The article has been fully improved by the authors. The topic of the article is fully in the scope of the journal.*

*Some minor points have to be fixed before publishing this article, that is why I recommend minor revisions.*

*p. 2 l.4 "to reasonable reproduce" I think it should be "to reproduce reasonably".*

*p. 3 l.2 "planning and prevention?" I think that the interrogation mark should be removed.*

*p.3 Concerning what is described from line 14 to line 27 which is very interesting, it might be nice to have a figure which summarises the methodology which is given here.*

*p.4 l.20 "is The Gen...": "The" should be changed into "the".*

*p.5 l.15 "... in areal means are likely, ..." I do not understand this sentence, I think that something is missing. It needs to be clarified.*

*p.7 l.4 "1D Mike 11 ..." a reference should be given concerning this hydraulic modelling tool.*

*p.7 l.28 "a uniform ..." should be changed into "an uniform ...".*

*p.8 l.17 "reason to selecting ..." I think it should be "reason to select ...".*

*p.9 l.26-27 "in Fig. 1 and 2 and Table1 ..." should be changed into "in Figs. 1 and 2 and 2 and Table 1 ...".*

*p.11 l.20 There is a mix between the physically based/mathematical model and the numerical method which is not very easy to understand. Moreover there is an error in the mass equation. So I recommend to write the mathematical model first and then to say few words on the way they are solved. Which should give :*

$\partial h/ \partial t +\partial Qx / \partial x +\partial Qy/ \partial y=0$

$\partial Qx/ \partial t = gA(Sx-n^2Qx|Q|/(R^{(4/3)}A^2) )$

$\partial Qy/ \partial t = gA(Sy-n^2Qy|Q|/(R^{(4/3)}A^2) )$

*where Q=(Qx,Qy), Sx (Sy) is the opposite of the slope in x (y) direction. It seems to be the kinematic wave model with an extra term.*

*p.11 l.23 "where Qis ..." into "where Q is".*

*p.11 l.26 "This model formulation is computationally more efficient than a full 2D dynamic model ..." it should be told why and by which factor.*

*p.13 l.33 "... given to predicting ..." I think it should be "... given to predict ...".*

*p.17 l.3 "The mean rainfall multiplier ... were ..." it should be plural on multiplier or singular on were.*

*p.17 l.8 "... for the these ..." the should be removed.*

*p.19 l.1 "the one-dimensional HEC-RAS in ..." should be changed into "the one-dimensional HEC-RAS model on ...". And a reference should be given concerning this modelling tool.*

*p.23 formula (A3) a point is missing at the end of the formula.*

*p.23 formula (A4) the same.*

*p.23 ( NWF) should be changed into (NWF).*

*p.23 l.10 and l.12 these are the only references for which "et al." is in italic. I recommend to put all Latin in italic (el al., i.e., ...).*

*p.24 "An initial ..." should be changed into "an initial ...".*

*p.24 l.2 "B1." should be changed into "B1:".*

*p.24 At the end of formula (B1) a point should be added.*

*p.24 l.4 "Eq. B2." into "Eq. B2:".*

*p.25 l.9 formula (B12) "...Ot," into "... Ot.".*

*p.25 l.16 formula (C1) a point should be added.*

*I recommend to center all the formulas.*

*All figures are fine accept figure 6 which is too small.*

**Reply to Referee #3**
We thank Referee #3 for her/his feedback and comments.

We appreciate the author's throughout revision and pointing out aspects for improvement of the manuscript. We went through all the reviewer's points and corrected accordingly, this include:

Regarding: *p.3 Concerning what is described from line 14 to line 27 which is very interesting, it might be nice to have a figure which summarises the methodology which is given here.*

- The incorporation of a new figure (Fig. 5) which will further increase the understanding of the method.

Regarding: *p.5 l.15 "... in areal means are likely, ..." I do not understand this sentence, I think that something is missing. It needs to be clarified.*

- We improved the text, see P.5 l.14-16.

Regarding:

*p.7 l.4 "1D Mike 11 ..." a reference should be given concerning this hydraulic modelling tool* and

*p.19 l.1 "the one-dimensional HEC-RAS in ..." should be changed into "the one-dimensional HEC-RAS model on ...". And a reference should be given concerning this modelling tool.*

- We added reference for 1D Mike 11 and HEC-RAS model.

Regarding: *p.11 l.20 There is a mix between the physically based/mathematical model and the numerical method which is not very easy to understand. Moreover there is an error in the mass equation.*

- We corrected the error in the mass equation and followed the advice of the reviewer for describing the momentum equation (see Eq. 1 and 2 in P.11 l.25-30 to P.12 l.1-11).

Regarding: *p.11 l.26 "This model formulation is computationally more efficient than a full 2D dynamic model ..." it should be told why and by which factor.*

- We added this information: P.12 l.6-9

Regarding: *p.23 l.10 and l.12 these are the only references for which "et al." is in italic. I recommend to put all Latin in italic (el al., i.e., ...).*

- We changed all Latin words to italic as suggested.

Regarding: *All figures are fine accept figure 6 which is too small.*

- We changed figure 6 (now 7) to show only the sensitive parameters, this was acknowledged in the figure caption and in the text (P.15 l.14-18), thus the redundant information from parameters that were no sensitive is not shown.

- In general all minor changes regarding improvements in the writing done by the reviewer were carried out.

[revised manuscript text omitted]

Comment [DF27]: This figure was added; accordingly, the number for the following figures and their reference in the text was changed

[Figure]

Figure 6 Precipitation (bars) and 100 class hydrographs chosen from the behavioural ones (black plots) for five sub-catchments upstream the floodplain. Predictive range of the 100% probability limits for all hydrographs simulations (grey shaded area) and rectangles representing the fuzzy set to allow for uncertainty for peak discharge and time of the peak for the sub-catchments of the Chiquito, Guacerique and Grande Rivers.

[Figure]

Figure 7 Prior (grey) and posterior (black outlined) relative frequency distribution for the for the most sensitive Rainfall-Runoff parameters: rainfall multiplier ($R$), rate of depletion ($m$), time factor ($t_d$) and the main channel roughness coefficient  ($n_{cu}$) for the Chiquito, Guacerique and Grande catchments (first, second and third row respectively).

[Figure]

Figure 8 Prior and posterior relative frequency distribution (grey and black outlined bars respectively) of the LISFLOOD-FP parameters (width factor, slope for the downstream boundary condition, channel roughness coefficient and floodplain roughness coefficient, $w_f$,

5    $b_c$, $n_c$ and $n_f$ respectively).

[Figure]

Figure 9 Prior and posterior relative frequency distribution (grey and black outlined bars respectively) of simulated maximum peak and time of the peak of input hydrographs for boundary conditions.

[Figure]

[Figure]

[Figure]

Figure 10 Performance of the model in predicting high-water marks, average ($d_{4-102}$), against predicted maximum peak discharge and two times of peak at Choluteca River (reference points 3 and 6 at Table 1) for non-behavioural simulations (grey dots), behavioural ones (black dots). Observed values and their limits of acceptability are plotted in continues and dotted vertical lines respectively.

[Figure]

Figure 11 Likelihood of high-water marks during the Mitch event, considering uncertainty in model parameters, model input and evaluation data to drive and constrain a combination of rainfall-runoff and hydraulic modelling tools.

[Figure]

Figure 12 Likelihood of inundated area during the Mitch event on 30–31 October 1998,
considering uncertainty in model parameters, model input and evaluation data to drive and
constrain a combination of rainfall-runoff and hydraulic modelling tools. The deterministic
flood extent was obtained by digitalisation of the flood extend in JICA (2002).